# Simulate Time-integrated Coarse-grained Molecular Dynamics with Multi-scale Graph Networks

**Xiang Fu**[*][†]
*Massachusetts Institute of Technology*

*xiangfu@csail.mit.edu*

**Tian Xie**[*][‡]
*Microsoft Research*

*tianxie@microsoft.com*

**Nathan J. Rebello**
*Massachusetts Institute of Technology*

*nrebello@mit.edu*

**Bradley D. Olsen**
*Massachusetts Institute of Technology*

*bdolsen@mit.edu*

**Tommi Jaakkola**[†]
*Massachusetts Institute of Technology*

*tommi@csail.mit.edu*

**Reviewed on OpenReview:** *https://openreview.net/forum?id=y8RZoPjEUl*

## Abstract

Molecular dynamics (MD) simulation is essential for various scientific domains but computationally expensive. Learning-based force fields have made significant progress in accelerating ab-initio MD simulation but are not fast enough for many real-world applications due to slow inference for large systems and small time steps (femtosecond-level). We aim to address these challenges by learning a multi-scale graph neural network that directly simulates coarse-grained MD with a very large time step (nanosecond-level) and a novel refinement module based on diffusion models to mitigate simulation instability. The effectiveness of our method is demonstrated in two complex systems: single-chain coarse-grained polymers and multi-component Li-ion polymer electrolytes. For evaluation, we simulate trajectories much longer than the training trajectories for systems with different chemical compositions that the model is not trained on. Structural and dynamical properties can be accurately recovered at several orders of magnitude higher speed than classical force fields by getting out of the femtosecond regime.

## 1 Introduction

Molecular dynamics (MD) simulation is widely used for studying physical and biological systems. However, its high computational cost restricts its utility for large and complex systems like polymers and proteins, which often require long simulations of nanoseconds to microseconds. While machine learning (ML) force fields (Behler & Parrinello, 2007; Unke et al., 2021b) have made significant progress in accelerating ab-initio MD simulations, their suitability for long-time simulations of large-scale systems is limited by several factors. Firstly, each force computation requires inference through the ML model. Although a single step with the ML model is much faster than ab-initio calculations, it is still slower than classical force fields. Secondly, simulating MD with a force field requires a very short time integration step at the femtosecond level to ensure stability and accuracy. This, combined with the slow single-step inference speed, results in extremely high

---

[*]Equal contribution.
[†]Correspondence to Xiang Fu and Tommi Jaakkola: xiangfu@csail.mit.edu, tommi@csail.mit.edu.
[‡]Work done at Massachusetts Institute of Technology.

costs for long-time simulations of large systems. Lastly, past research has shown that ML force fields are susceptible to instability in long-time simulations (Unke et al., 2021b; Stocker et al., 2022; Fu et al., 2022).

In many MD applications, we are interested in structural and dynamical properties at a large time/length scale. For example, the high-frequency vibrational motion of molecular bonds (femtosecond-level) is irrelevant when studying ion transport in battery materials (nanosecond-level). Computing the ion transport properties also does not require full atomic information, and a coarse-grained representation of the atomic environment suffices. To accelerate molecular dynamics simulations under large spatiotemporal scales, it is essential to develop a model that (1) can simulate beyond the femtosecond time step regime; (2) can simulate at a coarse-grained resolution that preserves properties of interest; (3) remains stable during long-time simulations.

To address these challenges, this paper proposes a fully-differentiable multi-scale graph neural network (GNN) model that directly simulates coarse-grained MD with a large time step (up to nanosecond-level) without computing or integrating forces. Our model includes an embedding GNN that learns transferrable coarse-grained bead embedding at the fine-grained level and a dynamics GNN that learns to simulate coarse-grained time-integrated dynamics at the coarse-grained level. Additionally, a refinement module based on diffusion models (Song & Ermon, 2019; Ho et al., 2020; Song et al., 2020; Shi et al., 2021) to resolve instability issues that may arise during long-time simulations. Although each step of the ML simulator is slower than classical force fields, the learned model can operate at spatiotemporal scales unreachable by classical methods, which leads to a dramatic boost in efficiency (summarized in Table 1).

The efficiency of our proposed method makes it particularly useful in high-throughput screening settings and enables us to conduct experiments on large-scale systems that are computationally prohibitive for existing ML force fields. Our evaluation protocol requires the simulation of thousands of atoms for billions of force field steps, which would take a typical ML (CG) force field months to simulate, whereas our model can complete within a few hours. In two realistic systems: (1) single-chain coarse-grained polymers in implicit solvent (Webb et al., 2020) and (2) multi-component Li-ion polymer electrolytes (Xie et al., 2022), We demonstrate that our model can (1) generalize to chemical compositions unseen during training; (2) efficiently simulate much longer trajectories than the training data; (3) successfully predicts various equilibrium and dynamical properties with a $10^3 \sim 10^4$ wall-clock speedup compared to the fastest classical force field implementations.

## 2 Related Work

Machine learning force fields (Unke et al., 2021b) aim to replace expensive quantum-mechanical calculations by fitting the potential energy surface from observed configurations with force and energy labels while being more efficient. An extensive series of research ranging from kernel-based methods to graph neural networks (Behler & Parrinello, 2007; Khorshidi & Peterson, 2016; Smith et al., 2017; Artrith et al., 2017; Chmiela et al., 2017; 2018; Zhang et al., 2018a;c; Thomas et al., 2018; Jia et al., 2020; Gasteiger et al., 2020; Schoenholz & Cubuk, 2020; Noé et al., 2020; Doerr et al., 2021; Kovács et al., 2021; Satorras et al., 2021; Unke et al., 2021a; Park et al., 2021; Thölke & De Fabritiis, 2021; Gasteiger et al., 2021; Friederich et al., 2021; Li et al., 2022; Batzner et al., 2022; Takamoto et al., 2022) has shown ML force fields can attain incredible accuracy in predicting energy and forces for a variety of systems. However, limited by slow inference and small time step sizes, simulation with large spatiotemporal scales is still inaccessible through ML force fields. This paper instead aims to learn a direct surrogate model at the CG level that bypasses force computation and enables efficient large-scale simulation.

Coarse-grained (CG) force fields (Marrink et al., 2007; Brini et al., 2013; Kmiecik et al., 2016) have been developed to extend the time and length scales accessible to molecular dynamics (MD) simulations while sacrificing some accuracy for computational efficiency. Despite this trade-off, these models have shown great success in simulating diverse biological and physical systems (Sharma et al., 2021; Webb et al., 2020). Machine learning methods have been used to learn CG mappings (Wang & Gómez-Bombarelli, 2019; Kempfer et al., 2019) and CG force fields through various approaches (Dequidt & Solano Canchaya, 2015; Wang et al., 2019; Husic et al., 2020; Greener & Jones, 2021; Chennakesavalu et al., 2022; Nkepsu Mbitou et al., 2022; Arts et al., 2023). However, A CG model that can accurately recover long-time properties is not known for every system (e.g., the Li-ion polymer electrolytes considered in this paper). Moreover, existing coarse-grained force fields for MD simulations are still limited by the femtosecond-level time step requirement for force

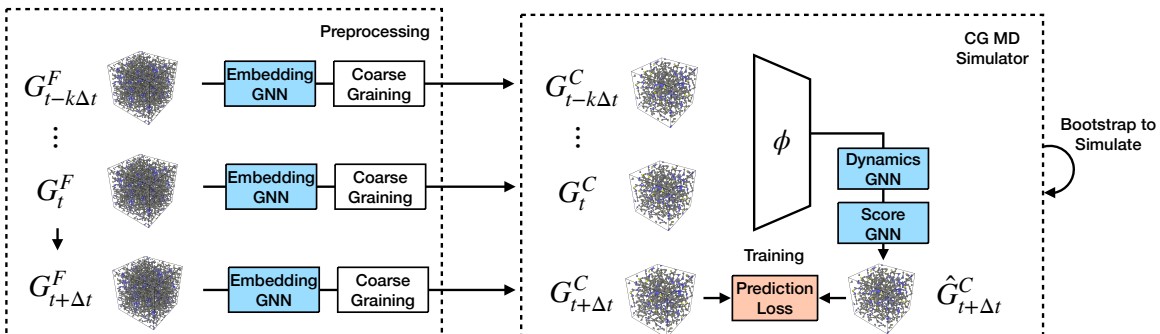

Figure 1: Learning time-integrated CG MD with multi-scale graph neural networks. Trainable modules are colored in blue. The loss term is colored in red. The preprocessing steps embed and coarse-grain an MD system to a coarse-level graph. The CG MD simulator processes historical information using the featurizer $\phi$, after which the Dynamics GNN predicts the next-step positions. A Score GNN is applied to refine the prediction. We bootstrap the CG MD simulator at evaluation time to make auto-regressive simulations.

integration. Our approach uses a general-purpose graph clustering algorithm for coarse-graining. It bypasses force integration to use much longer time steps with extreme coarse-graining while preserving key long-time structural and dynamical properties.

Enhanced sampling methods (Sugita & Okamoto, 1999; Laio & Parrinello, 2002; Barducci et al., 2008; Valsson & Parrinello, 2014; Torrie & Valleau, 1977) have proven successful in accelerating sampling of transition between metastable states in complex system dynamics such as protein folding (Bernardi et al., 2015; Schneider et al., 2017; Sultan et al., 2018; Yang et al., 2019). However, these methods rely on identifying collective variables (CVs) specific to each system, which can be challenging. Furthermore, they lack an explicit notion of time, making it difficult to estimate dynamical properties (Laio et al., 2005; Stelzl & Hummer, 2017). More recently, machine learning generative modeling approaches, such as Boltzmann generators, have enabled fast sampling of equilibrium states across phases (Noé et al., 2019; Mahmoud et al., 2022; Sidky et al., 2020; Vlachas et al., 2021; Kaltenbach & Koutsourelakis, 2021; Klein et al., 2023). Despite previous efforts, generating dynamical trajectories that can be applied to a wide range of chemical compositions and to large-scale systems with thousands of atoms remains a significant challenge. Another relevant line of work uses dimensional reduction techniques for building Markov state models and discovering long timescale kinetics (Pérez-Hernández et al., 2013; Mardt et al., 2018; Noé et al., 2016; Klus et al., 2018; Wu et al., 2018; Wu & Noé, 2020; Noé et al., 2020; Vlachas et al., 2021), but they don't directly construct an efficient MD simulator in the reduced space. Our goal in this work is to develop a general-purpose model that can simulate coarse-grained MD and recover equilibrium and dynamical long-time ensemble properties without prior knowledge of CVs.

## 3 Learning Time-integrated Coarse-grained Molecular Dynamics

**Model Overview.** As depicted in Figure 1, the learned simulator predicts single-step time-integrated CG dynamics at time $t + \Delta t$ given the current CG state and $k$ historical CG states at $t, t - \Delta t, \ldots, t - k\Delta t$. Here $\Delta t$ is the time-integration step, which is significantly longer than that used in MD simulation with force fields. Given ground truth trajectories at atomic resolution, we adopt a 3-step multi-scale modeling approach:

1. *Embedding*: learning atom embeddings at **fine level** using an Embedding GNN $\text{GN}_E$;

2. *Coarse-graining*: coarse-graining the system using graph clustering and constructing CG bead embedding from atom embedding learned at step 1;

3. *Dynamics* (and *Refinement*): learning time-integrated acceleration at **coarse level** using a Dynamics GNN $\text{GN}_D$. A Score GNN $\text{GN}_S$ is optionally learned to further refine the predicted structure.

**Representing MD trajectories as time series of graphs.** A ground truth MD simulation trajectory is represented as a time series of fine-level graphs $\{G_t^F\}$. The fine-level graph $G_t^F$ represents the MD state at time step $t$, and is defined as a tuple of nodes and edges $G_t^F = (V_t^F, E^F)$. Each node $\boldsymbol{v}_{i,t}^F \in V_t^F$ represents an atom [1], and each edge $\boldsymbol{e}_{i,j}^F \in E^F$ represents a chemical bond between the particles $\boldsymbol{v}_i^F$ and $\boldsymbol{v}_j^F$. The static fine-level graph $G^F = (V^F, E^F)$ describes all persistent features in an MD simulation, which include atom types, atom weights, and bond types. These persistent features are used to construct a time-invariant node representation that will be later used for CG bead embedding in our model. Applying the CG model to the fine-level graphs $\{G_t^F\}$ produces the time series of coarse-level graphs $\{G_t^C\}$, where the CG state $G_t^C$ is defined by the tuple $G_t^C = (V_t^C, E_t^C)$. Each node $\boldsymbol{v}_{m,t}^C \in V_t^C$ represents a CG bead, and each edge $\boldsymbol{e}_{m,n}^C \in E_t^C$ models an interaction between the CG beads $\boldsymbol{v}_{m,t}^C$ and $\boldsymbol{v}_{n,t}^C$. Since both non-bonded interactions and bonded interactions at the coarse level are significant for dynamics modeling, the edge set $E_t^C$ contains both CG-level bonds and radius cut-off edges constructed from the CG bead coordinates at time $t$.

**Graph neural networks.** A graph neural network takes graphs $G = (V, E)$ with node/edge features as inputs and processes a latent graph $G^h = (V^h, E^h)$ with latent node/edge representation through several layers of learned message passing. In this paper, we adopt the ENCODER-PROCESSOR-DECODER architecture (Sanchez-Gonzalez et al., 2020) for all GNNs, which inputs featurized graph and outputs a vector for each node. A forward pass through the GNN follows three steps: (1) The ENCODER contains a node multi-layer perceptron (MLP) that is independently applied to each node and an edge MLP that is independently applied to each edge to produce encoded node/edge features. (2) The PROCESSOR is composed of several layers of directed message-passing layers. It generates a sequence of updated latent graphs and outputs the final latent graph. The message-passing layers allow information to propagate between neighboring nodes/edges. (3) The DECODER is a node-wise MLP that is independently applied to the node features obtained from message passing to produce the outputs of a specified dimensionality. We refer interested readers to Sanchez-Gonzalez et al. 2020 for more details on the GNN architecture.

**Learning CG bead type embeddings with Embedding GNN.** Each node in the static fine-level graph $\boldsymbol{v}_i^F = [\boldsymbol{a}_i^F, w_i^F]$ is represented with (1) a learnable type embedding $\boldsymbol{a}_i^F$ that is fixed for a given atomic number and (2) a scalar weight $w_i^F$ of the particle. The edge embedding is the sum of the type embedding of the two endpoints and a learnable bond type embedding: $\boldsymbol{e}_{i,j}^F = [\boldsymbol{a}_i^F + \boldsymbol{a}_j^F + \boldsymbol{a}_{i,j}^F]$. The bond type embedding $\boldsymbol{a}_{i,j}^F$ is also fixed for a given bond type. We input this graph $G^F$ to the Embedding GNN $GN_E$ that outputs node embeddings $\boldsymbol{v}_i^F = [\boldsymbol{c}_i^F]$, for all $\boldsymbol{v}_i^F \in V^F$. This learned node embedding contains no positional information and will be used in the next coarse-graining step for representing CG bead types. The Embedding GNN is trained end-to-end with Dynamics GNN and Score GNN.

**Graph-clustering CG model.** Our CG model assigns atoms to different atom groups using a graph partitioning algorithm (METIS (Karypis & Kumar, 1998)) over the fine-level graph. The METIS algorithm partitions atoms into groups of roughly equal sizes while minimizing the number of chemical bonds between atoms in different groups. It first progressively coarsens the graph and partitions the coarsened graph, then progressively expands the graph back to its original size while refining the partition to obtain the final atom group assignment. Each node in the fine-level graph is assigned a group number: $C(\boldsymbol{v}_{i,t}^F) \in \{1, \ldots, M\}$, for all $\boldsymbol{v}_{i,t}^F \in V_t^F$. Here $M$ is the number of atom groups (CG beads). The graph partitioning is applied to the bond graph which is invariant through time as we study equilibrium state dynamics. This coarse-graining approach ensures that two atoms grouped into the same CG bead will never be far away from each other since they are connected by a path of bonds.

**Construct CG graph states.** We represent each fine-level node $\boldsymbol{v}_{i,t}^F \in V_t^F$ with $\boldsymbol{v}_{i,t}^F = [\boldsymbol{c}_i^F, w_i^F, \boldsymbol{x}_{i,t}^F]$, where $\boldsymbol{c}_i^F$ is the learned node type embedding from $GN_E$, $w_i^F$ is the weight, and $\boldsymbol{x}_{i,t}^F$ is the position. We can then obtain the coarse-level graph $G_t^C = (V_t^C, E_t^C)$ by grouping atoms in the same group into a CG bead. Denote the set of fine-level atoms with group number $m$ as $C_m = \{i : C(\boldsymbol{v}_{i,t}^F) = m\}$. The representation of the CG

---

[1]If the ground truth MD simulation already uses a CG model, each node will correspond to a CG bead defined by the CG model, and correspondingly the set of edges will be the set of all CG chemical bonds. For ease of presentation, in the rest of this paper, we refer to particles in the fine-level graph as "atoms".

bead $\boldsymbol{v}_{m,t}^C = [\boldsymbol{c}_m^C, w_m^C, \boldsymbol{x}_{m,t}^C]$ is defined as following:

$$
\boldsymbol{c}_m^C = \underset{C_m}{\mathtt{mean}}(\boldsymbol{c}_i^F) \equiv \frac{\sum_{i \in C_m} \boldsymbol{c}_i^F}{|C_m|}, \quad w_m^C = \underset{C_m}{\mathtt{sum}}(w_i^F) \equiv \sum_{i \in C_m} w_i^F, \quad \boldsymbol{x}_{m,t}^C = \underset{C_m}{\mathtt{CoM}}(\boldsymbol{x}_{i,t}^F, w_i^F) \equiv \frac{\sum_{i \in C_m} w_i^F \boldsymbol{x}_{i,t}^F}{\sum_{i \in C_m} w_i^F} \quad (1)
$$

The operators $\mathtt{mean}_{C_m}$ stands for taking the mean over $C_m$, $\mathtt{sum}_{C_m}$ stands for taking the sum over $C_m$, and $\mathtt{CoM}_{C_m}$ stands for taking the center of mass over $C_m$. Applying this grouping procedure for all atom groups $m \in \{1, \ldots, M\}$ creates the set of all CG nodes $V_t^C = \{\boldsymbol{v}_{m,t}^C : m \in \{1, \ldots, M\}\}$ for the coarse-level graph $G_t^C = (V_t^C, E_t^C)$.

We next construct the edges $E_t^C$. CG bonds are created for bonded atoms separated into different groups. We create a CG-bond $e_{m,n,t}^C$ if a chemical bond exists between a pair of atoms in group $C_m$ and group $C_n$. That is: $e_{m,n,t}^C \in E_t^C \impliedby \exists i \in C_m, j \in C_n$, such that $e_{i,j}^F \in E^F$. We further create radius cut-off edges by, for each CG bead, finding all neighboring beads within a pre-defined connectivity radius $r$ (with consideration to the simulation setup, e.g., periodic boundaries): $e_{m,n,t}^C \in E_t^C \impliedby \|\boldsymbol{x}_{m,t}^C - \boldsymbol{x}_{n,t}^C\| < r$. We set a large enough connectivity radius to capture significant interactions between all pairs of CG beads.

**Learning CG MD with Dynamics GNN.** The Dynamics GNN $\mathrm{GN}_D$ inputs a history-augmented coarse graph state $\phi(\{G_{t-i\Delta t}^C\}_{i=0}^k)$ and outputs a distribution of the time-averaged acceleration for each CG bead. The input node features of the featurized graph $\boldsymbol{v}_{m,t}^C = [\boldsymbol{c}_m^C, w_m^C, \{\dot{\boldsymbol{x}}_{m,t-i\Delta t}^C\}_{i=0}^k]$ include the CG bead type embedding $\boldsymbol{c}_m^C$, weight $w_m^C$, current and $k$-step history velocities $\{\dot{\boldsymbol{x}}_{m,t-i\Delta t}^C\}_{i=0}^{k-1}$. The input edge features $e_{m,n,t}^C = [\boldsymbol{x}_{m,t}^C - \boldsymbol{x}_{n,t}^C, \|\boldsymbol{x}_{m,t}^C - \boldsymbol{x}_{n,t}^C\|, \boldsymbol{c}_{m,n}^C]$ include displacement and distance between the two endpoints, and an embedding vector $\boldsymbol{c}_{m,n}^C$ indicating whether $e_{m,n,t}^C$ is a CG-bond or is constructed through radius cut-off. The output is a 3-dimensional Gaussian: $\mathrm{GN}_D(\phi(\{G_{t-i\Delta t}^C\}_{i=0}^k)) = \mathcal{N}(\boldsymbol{\mu}_t, \boldsymbol{\sigma}_t^2) = \{\mathcal{N}(\boldsymbol{\mu}_{m,t}, \boldsymbol{\sigma}_{m,t}^2) : \boldsymbol{v}_{m,t}^C \in V_t^C\}$ [2], where $\boldsymbol{\mu}_t$ and $\boldsymbol{\sigma}_t^2$ are the predicted mean and variance at time $t$, respectively. The training loss $L_{\mathrm{dyn}}$ for predicting the forward dynamics is thus the negative log-likelihood of the ground truth acceleration:

$$
L_{\mathrm{dyn}} = -\log \mathcal{N}(\ddot{\boldsymbol{x}}_t^C | \boldsymbol{\mu}_t, \boldsymbol{\sigma}_t^2)
$$

. The end-to-end training minimizes $L_{\mathrm{dyn}}$, which is only based on the single-step prediction of time-integrated acceleration. At inference time (for long simulation), the predicted acceleration $\hat{\ddot{\boldsymbol{x}}}_t \sim \mathcal{N}(\boldsymbol{\mu}_t, \boldsymbol{\sigma}_t^2)$ is sampled from the predicted Gaussian and integrated with a semi-implicit Euler integration to update the positions to the predicted positions $\hat{\boldsymbol{x}}_{t+\Delta t}$: $\hat{\dot{\boldsymbol{x}}}_{t+\Delta t} = \dot{\boldsymbol{x}}_t + \hat{\ddot{\boldsymbol{x}}}_t \Delta t$, $\hat{\boldsymbol{x}}_{t+\Delta t} = \boldsymbol{x}_t + \hat{\dot{\boldsymbol{x}}}_{t+\Delta t} \Delta t$.

**Learning to refine CG MD predictions with Score GNN.** We introduce the Score GNN, a score-based generative model (Song & Ermon, 2019) to resolve the stability issue of long simulation for complex systems. Following the noise conditional score network (NCSN) framework (Song & Ermon, 2019; Shi et al., 2021), the Score GNN is trained to output the gradients of the history-conditional log density (i.e., scores) given history state information and CG bead coordinates as input. An incorrect 3D configuration can be refined by iteratively applying the learned scores. With the refinement step, each forward simulation step follows a predict-then-refine procedure. The Dynamics GNN first predicts the (potentially erroneous) next-step positions, which the Score GNN refines to the final prediction $\hat{\boldsymbol{x}}_{t+\Delta t}$.

During training, the Score GNN is trained to denoise noisy CG bead positions to the correct positions. Let the noise levels be a sequence of positive scalars $\sigma_1, \ldots, \sigma_L$ satisfying $\sigma_1/\sigma_2 = \cdots = \sigma_{L-1}/\sigma_L > 1$. The noisy positions $\tilde{\boldsymbol{x}}_{t+\Delta t}^C$ is obtained by perturbing the ground truth positions with Gaussian noise: $\tilde{\boldsymbol{x}}_{t+\Delta t}^C = \boldsymbol{x}_{t+\Delta t}^C + \mathcal{N}(0, \sigma^2)$, where multiple levels of noise $\sigma \in \{\sigma_i\}_{i=1}^L$ are used. Due to coarse-graining, the coarse-level dynamics is non-Markovian (Klippenstein et al., 2021). Given the current state and historical information $\mathcal{H}$ that causes $\boldsymbol{x}_{t+\Delta t}^C$, the noisy positions follow the distribution with density: $p_\sigma(\tilde{\boldsymbol{x}}_{t+\Delta t}^C | \mathcal{H}) = \int p_{\mathrm{data}}(\boldsymbol{x}_{t+\Delta t}^C | \mathcal{H}) \mathcal{N}(\tilde{\boldsymbol{x}}_{t+\Delta t}^C | \boldsymbol{x}_{t+\Delta t}^C, \sigma^2 I) d\boldsymbol{x}_{t+\Delta t}^C$. The Score GNN outputs the gradients of the log density of particle coordinates that denoise the noisy particle positions $\tilde{\boldsymbol{x}}_{t+\Delta t}^C$ to the ground truth positions $\boldsymbol{x}_{t+\Delta t}^C$, conditional on the current state and historical information. That is, $\forall \sigma \in \{\sigma_i\}_{i=1}^L$:

---

[2] for ease of notation, in the rest of this paper we omit the node/edge indices when referring to every node/edge in the graph.

Table 1: Summary of datasets, simulation time scales, and evaluation observables. # atoms is the average number of atoms per system. CG level represents how many atoms are grouped into a single CG bead. GPU workflow for MD simulation is not always available. The simulation speed of classical MD on a GPU is usually 10x that of CPUs (Thompson et al., 2022), which would still be significantly slower than our method.

| Dataset | # atoms | CG level | Training traj. | Testing traj. | MD $\Delta t$ | Ours $\Delta t$ | MD cost | Ours cost | Key observable |
|---|---|---|---|---|---|---|---|---|---|
| Single-chain polymers | 890 | 100 | $100 \times 50\text{k } \tau$ | $40 \times 5\text{M } \tau$ | $0.01 \tau$ | $5 \tau$ | 32.7 CPU hrs | 0.10 GPU hrs | Radius of gyration |
| Li-ion polymer electrolytes | 6025 | 7 | $500 \times 5$ ns | $50 \times 50$ ns | 0.5 fs | $2 \cdot 10^5$ fs | 867.3 CPU hrs | 0.76 GPU hrs | Ion diffusivity |

$$
\begin{aligned}
\text{GN}_S([\boldsymbol{v}_t^{Ch}, \tilde{\boldsymbol{x}}_{t+\Delta t}^C]) &= \nabla_{\tilde{\boldsymbol{x}}_{t+\Delta t}^C} \log p_\sigma(\tilde{\boldsymbol{x}}_{t+\Delta t}^C | \boldsymbol{v}_t^{Ch}) \cdot \sigma \\
&\approx \nabla_{\tilde{\boldsymbol{x}}_{t+\Delta t}^C} \log p_\sigma(\tilde{\boldsymbol{x}}_{t+\Delta t}^C | \mathcal{H}) \cdot \sigma \qquad\qquad (*)\\
&= \mathbb{E}_{p(\boldsymbol{x}_{t+\Delta t}^C | \mathcal{H})}[(\boldsymbol{x}_{t+\Delta t}^C - \tilde{\boldsymbol{x}}_{t+\Delta t}^C)]/\sigma
\end{aligned}
$$

Note that in step $(*)$ we are approximating $\mathcal{H}$ with learned latent node embeddings $\boldsymbol{v}_t^{Ch}$, which is the last-layer hidden representation output by the Dynamics GNN. $\boldsymbol{v}_t^{Ch}$ contains current state and historical information. The training loss for $\text{GN}_S$ is:

$$
L_{\text{score}} = \frac{1}{2L} \sum_{i=1}^{L} \lambda(\sigma_i) \mathbb{E}_{\text{data}(\boldsymbol{x}_{t+\Delta t}^C | \mathcal{H})} \mathbb{E}_{p_{\sigma_i}(\tilde{\boldsymbol{x}}_{t+\Delta t}^C | \boldsymbol{x}_{t+\Delta t}^C)} \left[ \left\| \frac{\text{GN}_S([\boldsymbol{v}_t^{Ch}, \tilde{\boldsymbol{x}}_{t+\Delta t}^C])}{\sigma_i} - \frac{\boldsymbol{x}_{t+\Delta t}^C - \tilde{\boldsymbol{x}}_{t+\Delta t}^C}{\sigma_i^2} \right\|_2^2 \right] \qquad (2)
$$

where $\lambda(\sigma_i) = \sigma_i^2$ is a weighting coefficient that balances the losses at different noise levels. During training, the first expectation in Equation (2) is obtained by sampling training data from the empirical distribution, and the second expectation is obtained by sampling the Gaussian noise at different levels.

With the Score GNN, training is still end-to-end by jointly optimizing the dynamics loss and score loss: $L = L_{\text{dyn}} + L_{\text{score}}$. At inference time, the refinement starts from the predicted positions from Dynamics GNN. We use annealed Langevin dynamics, which is commonly used in previous work (Song & Ermon, 2019), to iteratively apply the learned scores to gradually refine the particle positions with a decreasing noise term. The predict-then-refine procedure can be repeated to simulate complex systems for long time horizons stably.

## 4 Experiments

Our experiments aim to substantiate our model's proficiency in effectively simulating coarse-grained, time-integrated dynamics for two large-scale atomic systems, which are concisely presented in Table 1. It is important to note that the primary objective of learning MD simulations is not the precise recovery of atomic states based on initial conditions. Rather, the focus lies in deriving macroscopic observables from MD trajectories to characterize the properties of biological and physical systems. This underlying motivation also drives the adoption of coarse-grained methodologies in previous studies as well as the present paper. To evaluate our simulation's performance, we compute essential macroscopic observables from the learned simulation and compare them to the reference data. Our model is contrasted with supervised learning baselines trained to directly predict observables from atomic structures. Our task presents a considerable challenge, as it demands generalization to various chemical compositions and extended time scales. As demonstrated in Table 1, for both datasets, the observables we aim to obtain necessitate lengthy simulations for estimation (length of testing trajectories), while the training trajectories are substantially shorter and inadequate for reliably extracting key observables. An ablation study is included in Appendix A. Further details on the simulation, observables, parameters, and baselines are included in Appendix B and Appendix C.

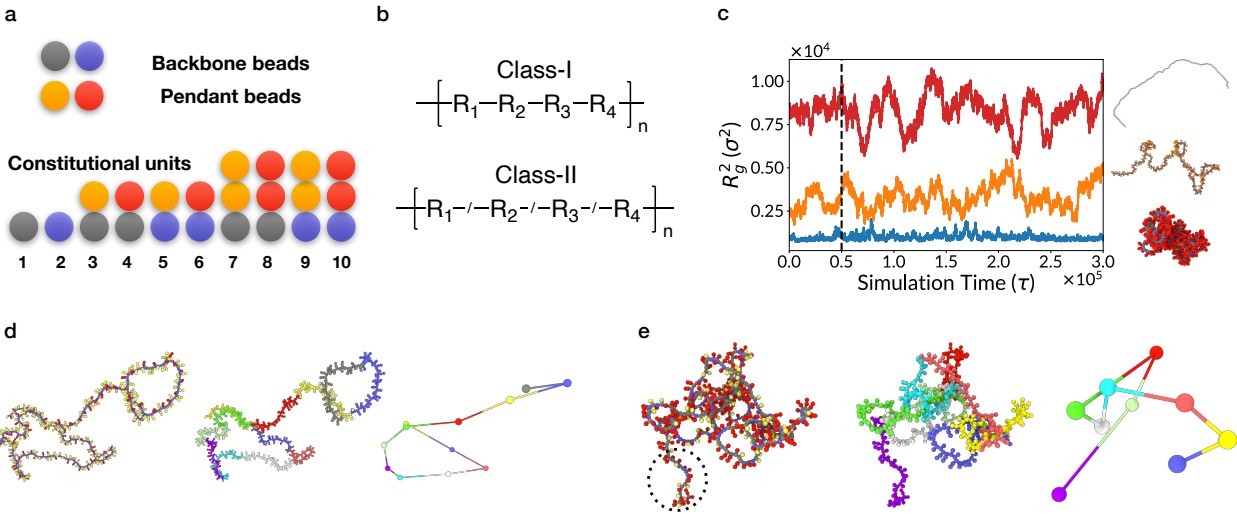

Figure 2: (a) All polymer chains are composed of 400 constitutional units, which are formed by four types of CG beads. (b) Class-I polymers are used for training, while class-II polymers are used for testing. The structure variation requires the model to learn generalizable dynamics. (c) $R_g^2$ for the three training polymers with smallest, median, and largest $\langle R_g^2 \rangle$, over a 300k $\tau$ period. Our training trajectories are 50k $\tau$ long (black dashed line), while we use 5M $\tau$ long trajectories for evaluation. (d) The coarse-graining process of a class-I polymer used for training. The first illustration shows the original polymer, the second illustration shows the CG assignment from the graph clustering algorithm (where beads belonging to the same super CG bead have the same color), and the third illustration shows the final CG configuration. More details on the CG process can be found at Equation (1). (e) The coarse-graining process of a class-II polymer used for testing, with similar illustrations to (d). We note the irregular structure in the circled area in the polymer illustration, which is a feature of class-II random polymers.

## 4.1 Single-chain Coarse-grained Polymers

In our first experiment, we focus on simulating single-chain coarse-grained polymers[3] within an implicit solvent. We employ the polymers introduced in Webb et al. 2020, where all polymers are composed of ten types of constitutional units that are formed by four types of CG beads (as shown in Figure 2 (a)). The interaction of the CG beads is described by a summation of bond, angle, and dihedral potentials, along with non-bonded interaction potential (detailed in Equation (3)). We train on regular copolymers with an exact repeat pattern of four CUs (class-I) and test on random polymers constructed from four CUs (class-II). This distribution shift (Figure 2 (b)) poses challenges to the generalization capability of ML models.

**Dataset and observables.** We train our model on 100 short class-I MD trajectories (with ten trajectories for validation) of 50k $\tau$, which are not sufficiently long for observable calculations. For evaluation, we use 40 testing class-II polymers using trajectories of 5M $\tau$ (100x longer). All polymers are randomly sampled and simulated under LJ units with a time-integration of 0.01 $\tau$. Each polymer contains 890 beads on average. A time step of $\Delta t = 5\tau$ is used for our model, so every step of the learned simulator models the integrated dynamics of 500 steps of the classical CG force field. We use the learned simulator at test time to generate 5M $\tau$ trajectories and compute properties. We coarse grain every 100 CG beads into a super CG bead. Examples of the coarse-graining process are illustrated in Figure 2 (d,e).

The squared radius of gyration $(R_g^2)$ is a property practically related to the rheological behavior of polymers in solution and polymer compactness that are useful for polymer design and has been the focus of several previous studies (Upadhya et al., 2019; Webb et al., 2020). Reliable estimation of $R_g^2$ statistics requires

---

[3]The classical MD definition is already for a coarse-grained system. To avoid confusion, in the rest of this paper, we only use "coarse-grained" when we are referring to the polymers after our graph partitioning-based coarse-graining approach.

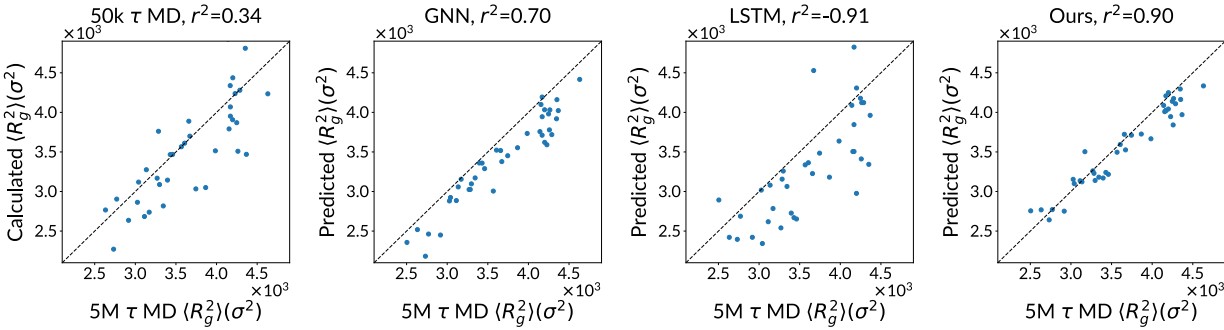

Figure 3: $\langle R_g^2 \rangle$ estimation performance using different methods. From left to right: (1) mean $R_g^2$ from 50k $\tau$ classical CG MD; (2) supervised learning GNN; (3) supervised learning LSTM; (4) our learned simulator.

sampling long MD simulation, while short simulation results in significant error. Figure 2 (c) shows how $R_g^2$ rapidly changes with time.

**Estimate $R_g^2$ using learned simulation.** In Figure 3 we demonstrate the experimental results on mean $R_g^2$ prediction across 40 testing polymers. The first panel shows the prediction made by short simulations of the same length as the training data will result in high-variance and poor recovery of mean $R_g^2$ due to insufficient simulation time. The second the third panels show the results of two baseline supervised learning (SL) models: one based on GNN and another based on long short-term memory (LSTM) (Hochreiter & Schmidhuber, 1997) networks. SL models input the polymer graph structure and directly predict the mean and variance of $R_g^2$ without learning the dynamics. These supervised learning models exhibit a systematic bias, possibly due to the distribution shift from training class-I polymers to testing class-II polymers. The fourth panel shows the results of our learned simulator, which demonstrates accurate recovery of $\langle R_g^2 \rangle$ and significantly outperforms the baseline models. This result shows that our model is able to learn dynamics transferrable to different chemical compositions and longer time horizons. Table 2 summarizes more performance metrics. Our method significantly outperforms the baseline models and estimation from short 50k $\tau$ MD trajectories in all metrics. In particular, the earth mover's distance (EMD) is a measure of distribution discrepancy. A lower EMD indicates a better match between the predicted $R_g^2$ distribution and the ground truth (Table 2). We also experiment with the existing learned simulator model (Sanchez-Gonzalez et al., 2020) without modifications but find it quickly diverges under the long-time simulation task setup.

**Distribution of $R_g^2$ and mean internal distance.** In Figure 4, we demonstrate our model's capability to reproduce $R_g^2$ distribution and mean internal distances for four selected testing polymers with small to large $\langle R_g^2 \rangle$. The first row shows our model can recover the distribution of $R_g^2$ very well, which is also quantitatively reflected by the low EMD in Table 2. The second row shows a good recovery of the mean internal distances across different length scales. While the learned simulator is only trained to make single-step predictions with short trajectories as training data, these statistics can be accurately recovered through a long simulation of 5M $\tau$ long, which is 500M steps of the original simulation. Our multi-scale modeling

Table 2: Performance for predicting $R_g^2$ statistics. $r^2$ and MAE are computed for $\langle R_g^2 \rangle$, and EMD is computed for $R_g^2$ distribution. To evaluate EMD, the SL models output a Gaussian distribution with the predicted mean and variance.

| Method | $r^2$ (↑) | MAE (↓) | EMD (↓) |
|---|---|---|---|
| Short MD, 50k $\tau$ | 0.34 | 349 | 4.10 |
| GNN, SL | 0.70 | 263 | 2.82 |
| LSTM, SL | -0.91 | 559 | 5.82 |
| Ours, 5M $\tau$ | **0.90** | **140** | **1.60** |

approach plays an important role in achieving this result: as every 100 CG beads are grouped into a single super CG bead, the features of each super CG bead must be well-represented to enable accurate dynamics under such a high degree of coarse-graining. The message passing of the embedding GNN, which happens at the fine-grained level, is capable of learning to represent the composition that is necessary for accurate

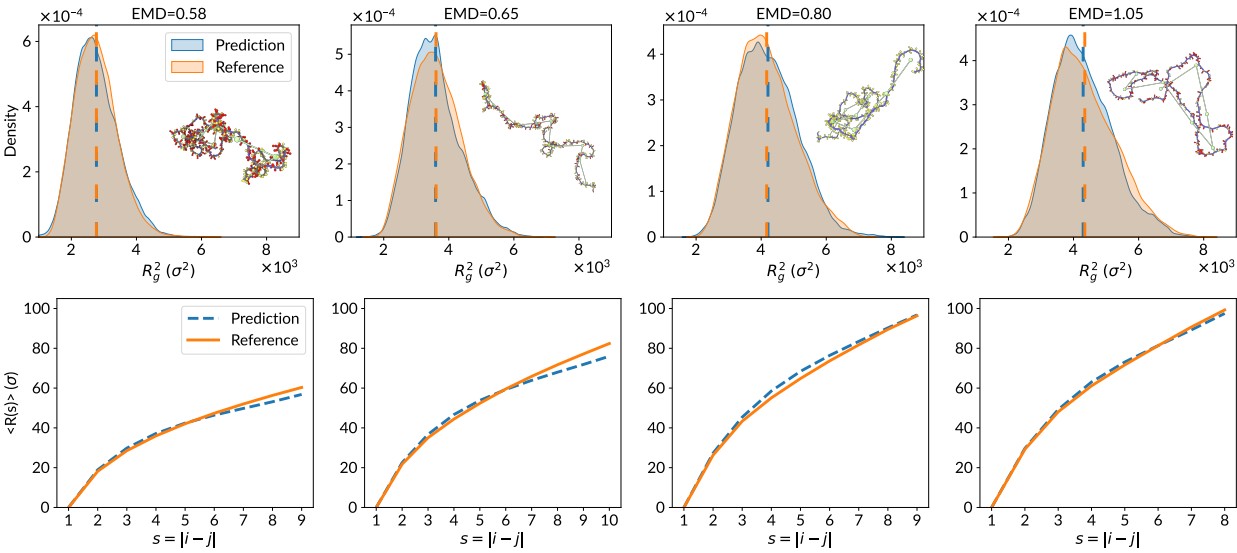

Figure 4: The distribution of $R_g^2$ from our learned simulation compared to the reference ground truth data; and the mean internal distance, respectively, for four example polymers with small to large $\langle R_g^2 \rangle$. In the first row, the dashed line annotates the mean $R_g^2$. Our model demonstrates accurate recovery of the distribution of $R_g^2$ and mean internal distance for various testing polymers unseen during training.

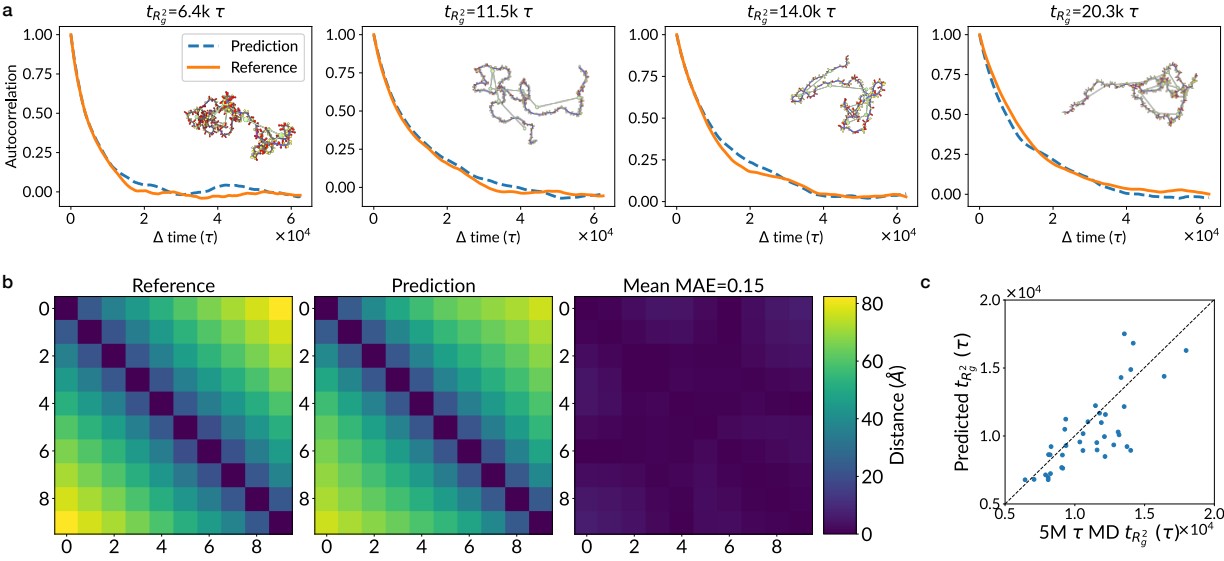

Figure 5: (a) Illustrations of four polymers and the autocorrelation function of $R_g^2$ for four polymers with the smallest, intermediate, and largest $R_g^2$ relaxation time. (b) The contact map for an example polymer produced by the reference simulation and the learned simulation. The third panel shows the absolute error. (c) Prediction performance of our model on the $R_g^2$ relaxation time.

dynamics, so the coarse-level dynamics GNN can carry out accurate simulations with orders of magnitude higher efficiency (as shown in Table 1).

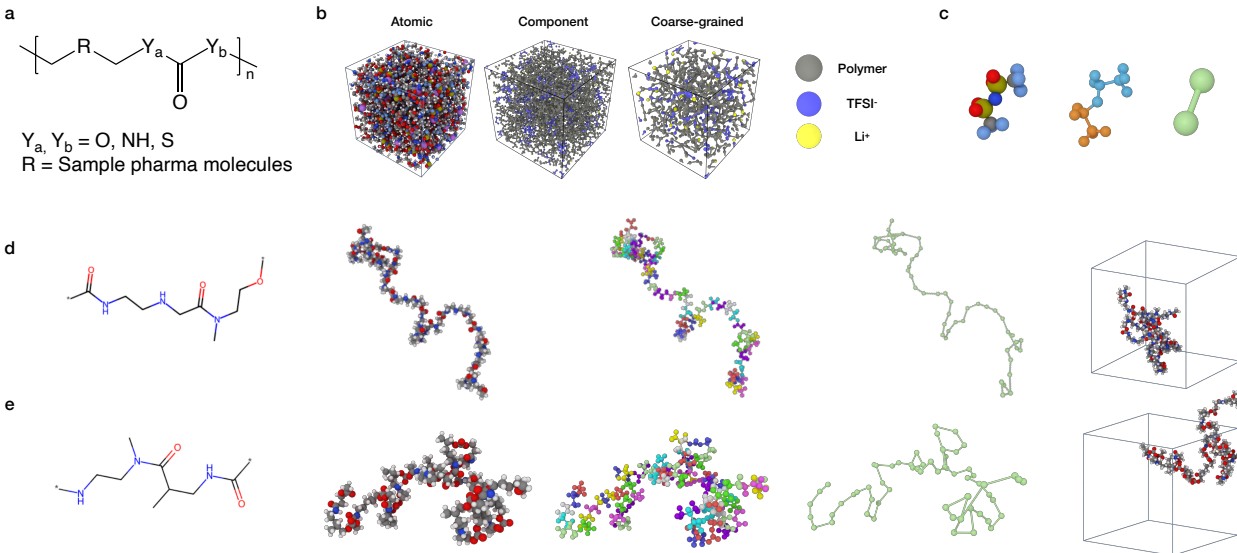

Figure 7: (a) The chemical space for the polymer in SPEs. Every trajectory contains a different type of polymer (around 10 chains, each chain has around 10 monomers), and the testing systems have distinct polymers from the training systems. (b) Example SPE illustrated in (1) full atomic structure; (2) colored coded by component; (3) coarse-grained structure. (c) The coarse-graining process for a TFSI-ion. (d) The coarse-graining process of a polymer chain in an example training SPE. From left to right: (1) the molecular graph of the monomer molecule, where connection points are marked with the symbol '*'; (2) the atomic structure of the polymer chain; (3) The CG assignment from the graph clustering algorithm; (4) The final coarse-grained configuration; (5) The relative scale of a single polymer chain in the periodic bounding box, with unwrapped coordinates. (e) The coarse-graining process of a polymer chain in an example testing SPE, with illustrations similar to (d).

**Relaxation time scale.** we show our learned simulator can capture the long-time dynamics using the autocorrelation function (ACF) of $R_g^2$ as our observable. The relaxation time is a long-time dynamical property significantly more challenging to estimate than mean $R_g^2$. It can not be obtained from many independent samples of polymer and requires very long simulations. Labels cannot be obtained from the training trajectories that are overly short, so the SL baselines are not applicable. Figure 5 (a) shows the ACF computed from our learned simulation matches the ground truth well for four polymers with the fastest, intermediate, and slowest decay of the $R_g^2$ ACF. We compute the relaxation time $t_{R_g^2}$ from the ACFs (details in Appendix B) and show the prediction of $t_{R_g^2}$ across all 40 testing polymers in Figure 5 (c). Recovery of $t_{R_g^2}$ shows that our model captures realistic dynamics rather than just the distribution of states. In Figure 5 (b), we visualize the contact map of an example polymer extracted from the reference simulation and our learned simulation. The contact map shows the pairwise distance between each pair of beads. Our model can accurately recover this detailed structural property.

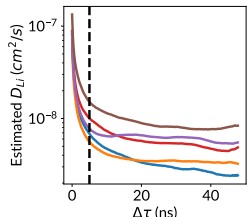

Figure 6: Estimation of Li-ion diffusivity with various trajectory lengths. The black dashed line annotates 5 ns (training trajectory length).

## 4.2 Multi-component Li-ion Polymer Electrolyte Systems

Solid polymer electrolytes (SPEs) are a type of amorphous material system that is promising in advancing Li-ion battery technology (Zhou et al., 2019) and is significantly more complex than single-chain polymers. Atomic-scale MD simulations have been an important tool in SPE studies (Webb et al., 2015; Xie et al., 2022), but are computationally expensive. The challenge of modeling these systems comes from (1) their

large size of thousands of atoms and the complexity of multiple components; (2) their amorphous nature and the slow convergence of key quantities, requiring long MD simulations on the order of 10 to 100 ns to estimate. For these reasons, screening a large chemical space with long simulations is extremely costly. Past work (Xie et al., 2022) has studied supervised learning approaches to predict ion transport properties but requires labels from long-time simulations to predict long-time properties accurately. In contrast, this work aims to predict long-time properties (50 ns) with short-time training data (5-ns trajectories) only by learning to simulate the dynamics that govern the system evolution.

We adopt the SPEs introduced in (Xie et al., 2022). Each MD trajectory is for a system with a distinct type of polymer. The polymer space is defined in Figure 7 (a), which contains monomers constructed with a large pharmaceutical database (Irwin & Shoichet, 2005). Each SPE system contains 6025 atoms on average. We train on 530 short MD trajectories of 5 ns (with 30 trajectories for validation) and evaluate on 50 testing SPE trajectories (with distinct polymers unseen during training) of 50 ns long. Our task is to predict the diffusivity of fast-moving ions – Li-ions and TFSI-ions ($D_{\mathrm{Li/TFSI}}$) – which is closely related to the conductivity of the battery material. One key challenge in simulating the Li-ion transport in SPEs is the slow relaxation process in amorphous polymers. Consequently, estimating the diffusivity of particles requires a very long simulation time to sample the dynamics, as shown in Figure 6. For accurate property estimation, we need trajectory lengths where $D_{\mathrm{Li}}$ converges. The 5-ns training trajectories are far from enough for extracting reliable Li-ion diffusivity, requiring the predictive method to generalize to longer time scales.

Our CG model groups every 7 bonded atoms into a CG bead using graph clustering. The coarse-graining process of the entire system is illustrated in Figure 7 (b); the coarse-graining process of a TFSI-ion is illustrated in Figure 7 (c); the coarse-graining process example polymer chains are illustrated in Figure 7 (d,e). Notably, we use a time-integration step of $\Delta t = 0.2$ ns, making each step of the learned simulator equivalent to $10^5$ steps in classical MD. Such a long time step will smooth out all high-frequency vibrational modes of motion while slow modes of motion are preserved. It enables dramatic speedup of simulation compared to force-field-based simulations with femtosecond-level time steps. Ablation studies on the hyperparameters are included in Appendix A.

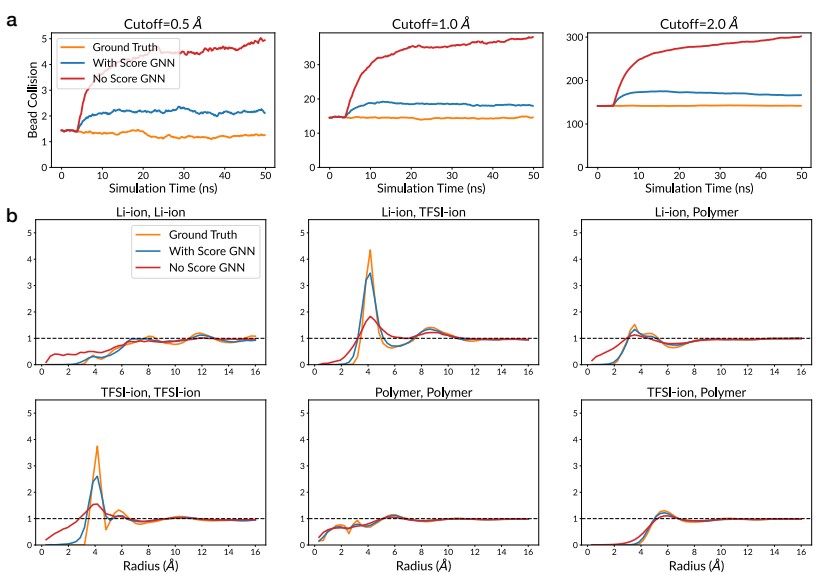

Figure 8: (a) Bead collision as a function of simulation time averaged over the 50 testing SPEs for all methods. The learned models start simulating at 4 ns. From left to right: we consider cut-off radii of 0.5 Å, 1.0 Å, and 2.0 Å. (b) RDFs of different types of particles in the SPE systems for our model with/without the Score GNN refinement and the ground truth MD simulation averaged over a 50-ns trajectory of an example SPE.

**Stability of learned simulation and Score GNN refinement.** Stability is a prerequisite for the accurate recovery of simulation ensemble properties. As a measure of stability, we consider a "bead collision" happens when two beads have a distance below a cut-off radius. At a selected cut-off radius, a realistic level of bead collision can be computed from the reference data. Figure 8 (a) demonstrates the number of bead collisions as a function of simulation time, averaged over all 50 test SPEs, for three different cut-off radii: 0.5 Å, 1.0 Å, and 2.0 Å. Simulation without Score GNN refinement suffers from a much higher level of bead collision, and the system becomes increasingly unphysical

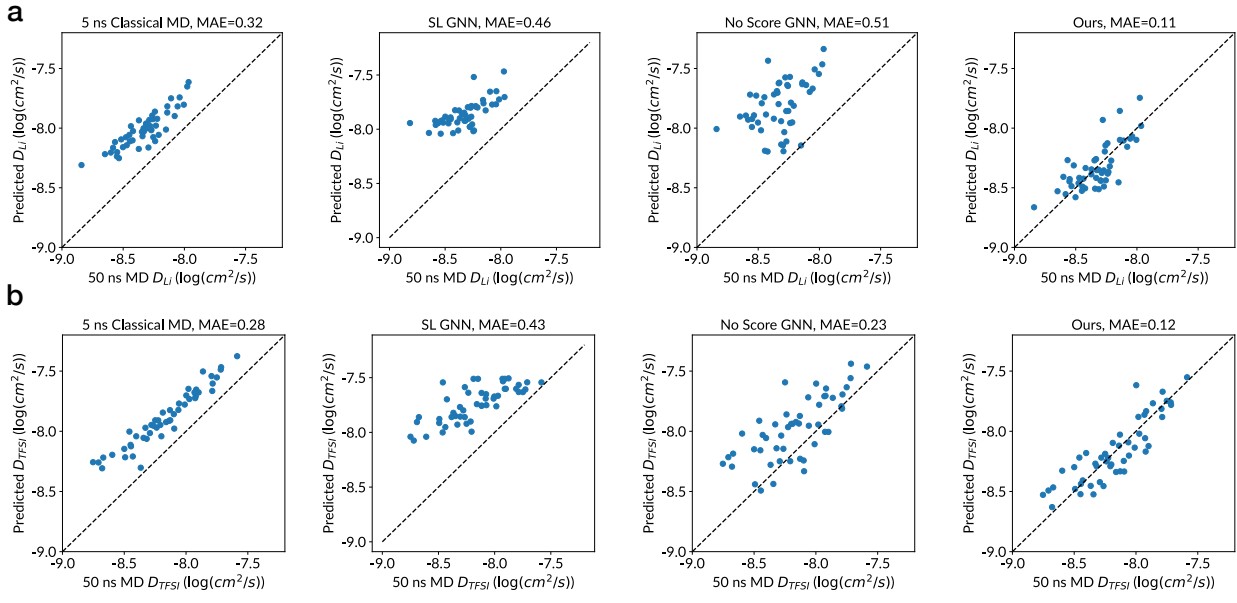

Figure 9: Li-ion diffusivity (a) and TFSI-ion diffusivity (b) estimation performance using different methods. From left to right: (1) 5-ns MD, which is the length of the training trajectories; (2) Supervised learning GNN model (Xie et al., 2022); (3) Our model without the Score GNN refinement; (4) Our full model.

as simulation proceeds. The Score GNN refinement significantly reduces collision to a level similar to the reference, consistently across all cut-off radii. Radial distribution functions (RDFs) are commonly used to describe the structural properties of a physical system (definition in Appendix B). In Figure 8 (b), we plot the RDFs averaged over the 50-ns simulation for an example testing SPE over beads from different components. Without the Score GNN refinement, we observe excessively high density at a near distance (which indicate abnormal bead collision and unphysical strctures) and missing radius peaks. With the Score GNN refinement, the learned simulation produces RDFs that can recover the reference's overall shape and radius peaks.

**Estimate ion transport properties.** We show the Li-ion/TFSI-ion diffusivity prediction performance of different models in Figure 9. The first column panels of Figure 9 (a,b) show the prediction from 5-ns classical MD, which is the length of the training trajectories, giving a poor estimation of Li-ion/TFSI-ion diffusivity. The second column panels show the results from the supervised learning GNN model implemented in (Xie et al., 2022) while fitting an exponential decay curve (detailed in Appendix C) to account for the convergence of $D_{\text{Li/TFSI}}$ (as shown in Figure 6) for the Li-ion/TFSI-ion diffusivity. Different from the experimental settings in (Xie et al., 2022), in our experiments, no 50-ns trajectory is available for training. We see the supervised model is not capable of recovering the Li-ion/TFSI-ion diffusivity with only short trajectories for training and an exponential decay estimation. The third column panels show the performance of our model but without the Score GNN refinement. Due to the unstable dynamics as shown in Figure 8, this model is not able to recover Li-ion/TFSI-ion diffusivity. The last column panels show the performance of our full model, which can simulate stably and accurately recover the 50-ns Li-ion/TFSI-ion diffusivity of unseen systems by training from short trajectories of 5 ns only, with several orders of magnitude higher efficiency (Table 1).

# 5 Discussion

We have developed a machine learning model for simulating coarse-grained MD using very large time-integration steps without integrating forces. The proposed model can recover key long-time statistics accurately while being several orders of magnitude faster than classical MD. The superior efficiency allows us to study large-scale systems that are computationally inaccessible to ML force fields and ML CG force

fields. The learned simulator is trained over short MD trajectories but can generalize well to simulate longer trajectories for novel unseen systems, making it highly suited for high-throughput screening settings. In two challenging and realistic applications, our model significantly outperforms supervised learning baseline methods and directly uses short-time MD for property estimation.

On the other hand, coarse-grained simulations necessarily entail trade-offs between efficiency and accuracy. Unlike force fields, which are conserved vector fields, our method cannot guarantee energy conservation or time reversibility. Additionally, the high degree of coarse-graining and very long time steps make the dynamics non-Markovian and appear stochastic, which poses difficulties for predicting system behavior (Klippenstein et al., 2021). Consequently, maintaining stability proves to be highly challenging for our model. To tackle the instability issue, this work introduces a novel refinement module based on diffusion models, for which the stabilization effect is verified through experiments. Extending sample efficient equivariant ML force field model architectures (Batzner et al., 2022) to our setting and active learning for collecting high-quality data (Vandermause et al., 2020) are some other promising directions for future development of time-integrated simulators.

The dynamics simplification introduced by spatial coarse-graining and temporal time integration is a two-edged sword: the simplified dynamics can be easier to learn and more efficient to simulate, but the lost information may lead to inaccurate dynamics and errors in property estimation. Therefore, optimal CG modeling is closely tied to the objective observables of the MD simulation. CG modeling techniques based on the essential dynamical information (Kmiecik et al., 2016; Souza et al., 2021; Zhang et al., 2018b; Wang & Gómez-Bombarelli, 2019; Li et al., 2020) is another important future direction in designing more effective CG MD simulation schemes.

## Acknowledgments

We pay tribute to Octavian-Eugen Ganea (1987-2022), a dear colleague and friend who we had many inspiring discussions over this work. We thank Wujie Wang, Zhenghao Wu, Tzyy-Shyang Lin, Gabriele Corso, Bowen Jing, Shangyuan Tong, Yilun Xu, Hannes Stärk, the rest of the TJ group members and anonymous TMLR reviewers for their helpful comments and suggestions. We thank Michael A. Webb for his advice and support on the single-chain coarse-grained polymer dataset. We thank Jacob Helwig for pointing out a typo in our paper. We gratefully thank the Community Resource for Innovation in Polymer Technology, a project supported by the National Science Foundation (NSF) Convergence Accelerator program (Convergence Accelerator Research 2134795), and MIT-GIST collaboration for support. The authors also acknowledge the MIT SuperCloud and the Lincoln Laboratory Supercomputing Center for providing HPC resources.

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

## A    Ablation Studies

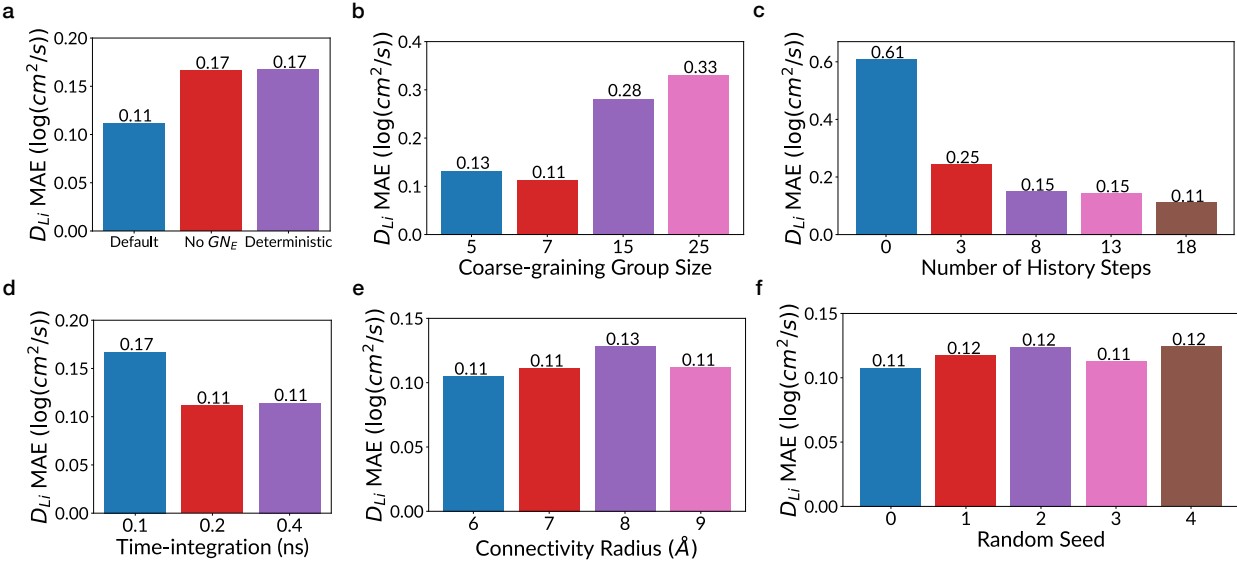

Figure 10: (a) Comparison of our default model, a model without the Embedding GNN at the fine-level graph, and a model that predicts deterministic acceleration for each particle. (b) Comparison of different coarse-graining atom group sizes. The atom group size is the number of atoms contained in each CG bead. Performance drops when the coarse-graining is too fine (5) or too coarse (15, 25). (c) Comparison of different history lengths for predicting the dynamics. We observe that longer history helps improve the model performance. (d) Comparison of different time-integration step lengths. A longer time integration removes high-frequency information and simplifies the dynamics, making it easier to learn. On the other hand, the long-time property of particle diffusivity does not require high-frequency information to estimate accurately. (e) Comparison of different connectivity radii when building the coarse-level graph. A radius of 6 Åis sufficient for modeling the SPE systems. (f) Comparison of different random seeds using the same model. Our model does stochastic rollouts, and the performance is robust to random seeds.

**Fine-level Embedding GNN** $GN_E$**.** (Figure 10 (a)) Our model uses a fine-level embedding GNN to learn CG bead embedding that enclose local structural information. We can remove this fine-level GNN and let the CG bead embedding be the mean over the atom type embedding in each atom group. As shown in Figure 10 (a), model performance significantly drops without the Embedding GNN.

**Stochastic dynamics prediction.** (Figure 10 (a)) We attempt to let our model output deterministic acceleration at each forward simulation step. However, the inherent uncertainty makes the model predict very small movement for all particles at every step, and the model performance significantly drops. The small movements lead to slow transport of particles, and causes underestimation of Li-ion diffusivity, as shown in Figure 11 (a). Such unrealistic dynamics also makes long simulation unstable. This is demonstrated in Figure 11 (b), which shows that a deterministic model has a higher number of bead collision (under a cutoff of 1.0 Å) with an increasing trend through time.

**Coarse-graining group size.** (Figure 10 (b)) We experimented with different coarse-graining group size. We observe that in terms of capturing particle diffusivity, using a group size of 7 outperforms finer (5) and coarser (15, 25) coarse-graining. We hypothesize that the finer system is hard to model accurately with limited training data, while the coarser system loses important information for accurate dynamics modeling. This result further suggests the optimal coarse-grained modeling should be conditional on the objective of MD simulation.

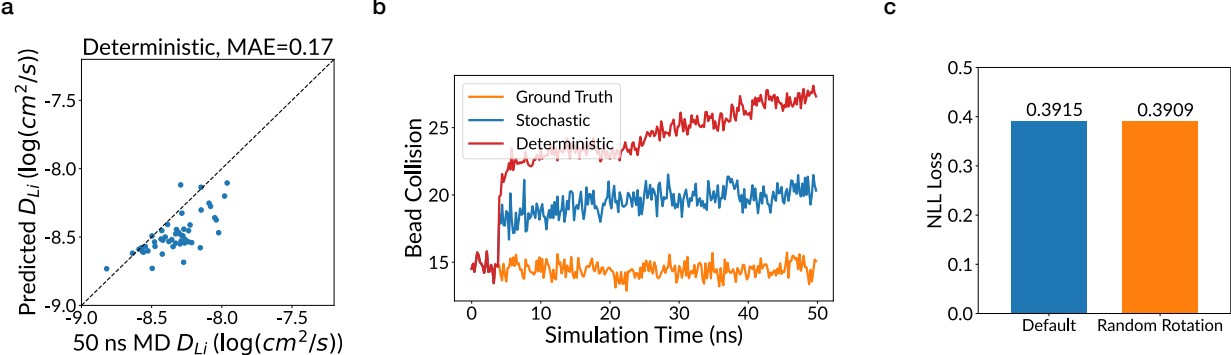

Figure 11: (a) Li-ion diffusivity estimation performance of the deterministic learned simulator. The model predicts small movements at every step, leading to slower ion transport and underestimation of Li-ion diffusivity. (b) Bead collision at 1 Å as a function of time, averaged over the 50 testing SPEs. the deterministic model's prediction becomes increasingly unphysical as simulation proceeds. (c) The negative log-likelihood (NLL) loss of model prediction on testing polymers. Model performance remains the same when the input structures are randomly rotated before being fed into the model.

**Use of historical information for prediction.** (Figure 10 (c)) The spatio-temporal coarse-graining introduces memory effects to the resulting dynamics. Our model uses historical information by utilizing $k$-step historical velocities in dynamics prediction, and we experimented with $k = 0, 3, 8, 13, 18$, under a time-integration of 0.2 ns per step. We observe that longer history as input leads to better performance.

**Time-integration step size.** (Figure 10 (d)) We experimented with different time-integration step size. We observe that a longer time-integration gives the best performance. We hypothesize this is due to the dynamics simplification effect of long time-integration, while the loss of high-frequency dynamics does not damage the estimation accuracy of particle diffusivity under long-time simulation.

**Connectivity radius.** (Figure 10 (e)) Our coarse-level graphs contain both CG-bonds and radius cut-off edges. The radius cut-off edges model non-bonded interactions. We experimented with radius 6, 7, 8, 9 Å, but found no significant performance difference. We conclude that a radius of 6 Åis sufficient for modeling the SPE non-bonded dynamics.

**Random seeds.** (Figure 10 (f)) As our model does stochastic simulation, we examine its robustness against the random seed. We rollout 5 times using the same model and observed no significant performance difference across random seeds.

**SE(3) Equivariance.** Some molecular systems obey symmetries in the form of invariance and equivariance. While translational equivariance is enforced by the proposed model, effective rotational equivariance is learned from data. The effective equivariance is verified by an experiment that applies random rotations at X, Y, and Z axes to input data. The time-integrated acceleration predicted by our model is accordingly rotated by the same amount (Appendix A, Figure 11). We present our model as a general approach that is applicable to diverse systems since some MD systems are not rotationally equivariant (e.g., when an external field such as an electric field is present) with favorable time efficiency. Extending the proposed framework to equivariant neural networks (Thomas et al., 2018; Satorras et al., 2021; Batzner et al., 2022; Gasteiger et al., 2020; 2021) is a promising future direction to improve data efficiency and generalization capability. Such extension will involve incorporating historical information in an equivariant way and improving the efficiency of equivariant GNNs for training and inference with large-scale systems and big dataset sizes.

We verify our learned simulator being rotationally equivariant by applying random rotations at X, Y, and Z axes to input data, and compute the dynamics prediction loss, before and after the random rotations. Figure 11 (c) shows that the model test time performance is unchanged irrespective of the random rotations. Therefore, our learned simulator is effectively rotational equivariant by learning from data.

# B  Dataset Details

**Single-chain polymer dataset.** The single-chain polymers are simulated using LAMMPS (Thompson et al., 2022), with the force-field parameters and chemical space defined in (Webb et al., 2020). For clarity, we present the potential terms below, while more details can be found in (Webb et al., 2020).

$$U(\boldsymbol{r}^N) = \sum_{\text{bonds}} U_{\text{vib}}(r_{ij}) + \sum_{\text{angles}} U_{\text{bend}}(\theta_{ijk})$$
$$+ \sum_{\text{dihedrals}} U_{\text{tors}}(\phi_{ijkl}) + \sum_{ij} U_{\text{nb}}(r_{ij}) \tag{3}$$

where $r_{ij}, \theta_{ijk}$, and $\phi_{ijkl}$ are internal distances, angles, and dihedrals, respectively. The individual potential terms are given by:

$$U_{\text{vib}}(r_{ij}) = -\frac{1}{2} K_{ij} (R_{ij}^{(0)})^2 \log \left[ 1 - \left( \frac{r_{ij}}{R_{ij}^{(0)}} \right)^2 \right]$$

$$U_{\text{bend}}(\theta_{ijk}) = K_{ijk} (\theta_{ijk} - \theta_{ijk}^{(0)})^2$$

$$U_{\text{tors}}(\phi_{ijkl}) = K_{ijkl} [1 + \cos \phi_{ijkl}]$$

$$U_{\text{nb}}(r_{ij}) = \begin{cases} 4\epsilon_{ij} \left[ \left( \frac{\sigma_{ij}}{r_{ij}} \right)^{12} - \left( \frac{\sigma_{ij}}{r_{ij}} \right)^6 \right] & \text{if } i,j \text{ bonded} \\ & \text{and } r_{ij} < 2^{1/6} \\ 4\epsilon_{ij} \left[ \left( \frac{\sigma_{ij}}{r_{ij}} \right)^9 - \left( \frac{\sigma_{ij}}{r_{ij}} \right)^6 \right] & \text{otherwise} \end{cases}$$

All simulations are done in reduced units with characteristic quantities $\sigma$ for distance and $\tau$ for time. Single-chain polymer dynamics in implicit solvent evolve according to the Langevin equation using the velocity-Verlet integration scheme. We use a time step of $0.01\tau$. Training trajectories are 50k $\tau$ after removing the initial trajectory for relaxation and are recorded every 5 $\tau$. Testing trajectories are 5M $\tau$ and are recorded every 500 $\tau$. The polymer interaction is described by the summation of bonded and non-bonded potential energy functions. We refer interested readers to (Webb et al., 2020) for more details on the simulation setup.

**Calculation of single-chain polymer properties.** Our main property of study, the squared radius of gyration $(R_g^2)$ is computed by:

$$R_g^2 = \left( \frac{\sum_{i \in V} m_i d_i^2}{\sum_{i \in V} m_i} \right)$$

where $V$ is the set of all nodes, $m_i$ is the mass of particle $i$, and $d_i$ is the distance from particle $i$ to the center of mass. The mean internal distance $\langle R(s) \rangle$ is the average distance between bead $i$ and bead $i+s$ on the single chain. It is computed as:

$$\langle R(s) \rangle = \sum_{i=1}^{M-s} \frac{\|\boldsymbol{x}_i - \boldsymbol{x}_{i+s}\|}{M-s}$$

Where $M$ is the total number of CG beads in the single chain. The relaxation time for $R_g^2$ is derived from the autocorrelation function (ACF), which is computed as:

$$\text{ACF}(y) = \frac{\langle R_g^2(t)^2 R_g^2(t+y)^2 \rangle_t - (\langle R_g^2(t)^2 \rangle_t)^2}{\langle R_g^2(t)^4 \rangle_t - (\langle R_g^2(t)^2 \rangle_t)^2}$$

Here $\langle \cdot \rangle_t$ stands for averaging over the entire trajectory. The relaxation time $t_{R_g^2}$ is computed by fitting an exponential function $f(y) = \exp(-t_{R_g^2} y)$ to the ACF, and the relaxation time is the time when the ACF decays to $1/e$. It is a highly dynamic long-time property decided by the polymer structure.

**SPE dataset.** The SPE systems are simulated using LAMMPS (Thompson et al., 2022) with the force-field parameters and chemical space defined in (Xie et al., 2022). The atomic interactions are described by the polymer consistent force-field (PCFF+) (Sun, 1994; Rigby et al., 1997). Polymer amorphous structures are sampled with a Monte Carlo algorithm and then mixed with 1.5 mol lithium bis-trifluoromethyl sulfonimide (LiTFSI) per kilogram of polymer. For all systems, there are 50 Li-ions and TFSI-ions in the simulation box, and each polymer chain has 150 atoms in the backbone. The training trajectories are 5 ns long after removing the initial equilibration, while the testing trajectories are 50 ns long. All trajectories are recorded after 5 ns MD equilibration. Each system is run in the canonical ensemble (nVT) at a temperature of 353K using a multi-timescale integrator with an outer time step of 2 fs for nonbonded interactions and an inner time step of 0.5 fs. We refer interested readers to (Xie et al., 2022) for more details on the simulation setup.

**Calculation of SPE properties.** The radial distribution function (RDF) describes the particle density as a function of distance from a reference particle. For a system with many distinct types of particles, we can compute RDF for particular types by counting the neighbor atoms of certain types at various radii and each time step, and average over the entire trajectory. The RDF for particle types $A, B$ at distance $r$ is computed as:

$$\text{RDF}_{A,B}(r) = \frac{d[n_{A,B}(r)]}{4\pi r^2 dr}$$

Here $n_{A,B}(r)$ is the number of particle pairs with types $A$ and $B$ and distance in $[r, r + dr)$, where $dr$ is a small bin size. We include more RDF results from the learned simulation in Figure 16.

Ion transport properties of SPEs are the topic of study for many previous works and require long-time simulation to estimate. In particular, our experiments focus on particle diffusivity. With a trajectory of time horizon $T$, the diffusivity $D$ of a particle is computed by:

$$D = \frac{\|\boldsymbol{x}_T - \boldsymbol{x}_0\|_2^2}{6T}$$

where $\boldsymbol{x}_T$ is the position at time $T$, and $\boldsymbol{x}_0$ is the position at time 0. The diffusivity of a type of particle (e.g., Li-ion) is then computed by averaging the particle diffusivity over all particles of that type.

## C Experimental Details

---
**Algorithm 1** annealed Langevin dynamics for structural refinement

---
1: **Input**: $\text{GN}_D$ prediction $\hat{\boldsymbol{x}}_{t+\Delta t}$, noise levels $\{\sigma_i\}_{i=1}^{L}$, denoising step size $\epsilon$, steps per noise level $T_\sigma$
2: **Output:** denoised positions $\bar{\boldsymbol{x}}_{t+\Delta t}$
3: Initialize $\bar{\boldsymbol{x}}_{t+\Delta t} = \hat{\boldsymbol{x}}_{t+\Delta t}$
4: **for** noise level $i = 1, \dots, L$ **do**
5:     **for** step $s = 1, \dots, T_\sigma$ **do**
6:         Sample positional noise $\boldsymbol{z}_s \sim \mathcal{N}(\boldsymbol{0}, \boldsymbol{I})$
7:         $\bar{\boldsymbol{x}}_{t+\Delta t} \leftarrow \bar{\boldsymbol{x}}_{t+\Delta t} + \epsilon \cdot \text{GN}_S(\bar{\boldsymbol{x}}_{t+\Delta t})/\sigma_i + \sqrt{2\epsilon}z_s$
8:     **end for**
9: **end for**

---

In our implementation, we use 2 hidden layers for all MLPs and 7 message-passing layers for all GNNs. The Embedding GNN $\text{GN}_E$ has a hidden size of 64, while the Dynamics and Score GNNs have a hidden size of 128. All activation functions in the neural networks are rectified linear units (ReLU). We train the model for 2 million steps. The network is optimized with an Adam optimizer with an initial learning rate of $2 \times 10^{-4}$, exponentially decayed to $2 \times 10^{-5}$ over the 2 million training steps. All models are trained and used for producing long simulations over a single RTX 2080 Ti GPU. In the single-chain polymer experiments, the fine-grained $R_g^2$ is not computable from the coarse-grained configuration. To obtain the fine-grained $R_g^2$ from CG simulations, We first compute the $R_g^2$ from the CG configuration and then use an MLP over the latent graph embedding of the dynamics GNN to predict the difference between the CG $R_g^2$ and FG $R_g^2$. We then predict the FG $R_g^2$ by adding the CG $R_g^2$ and the offset predicted by this MLP module. Since the simulation is stable, the refinement module is not used for the single-chain polymers. For the SPE systems, the refinement

module is a diffusion model with 20 noise levels ranging from [0.01, 10.]. At simulation time, the refinement is through an annealed Langevin dynamics of 10 steps per noise level and a step size of $5 \cdot 10^{-5}$. The refinement procedure is described in Algorithm 1.

**Single-chain polymer baseline models.** We conduct experiments with two supervised learning baseline models. The first one is a GNN model that takes the polymer chemical graph as input and outputs the mean and standard deviation of $R_g^2$. Each node in the polymer graph is a CG bead and each edge is a chemical bond. The GNN uses the ENCODER-PROCESSOR-DECODER with 10 message-passing layers to process the polymer graph to a latent graph with node/edge embeddings. The MLPs in the GNNs have 2 hidden layers with a size of 128. The node embeddings are then summed to get the graph embedding, which is then processed by a 2-layer MLP with the hidden size 256 to obtain the final prediction of mean and standard deviation of polymer $R_g^2$. For accurate $R_g^2$ prediction of the fine-grained state in the single-chain polymer experiment, we use another neural network that inputs the coarse-level latent graph representation to fit the residual of $R_g^2$ computed from the coarse-level states (as opposed to the fine-level states). The second LSTM baseline model takes the 1D polymer chain structure as its input and replaces the GNN encoder with a 2-layer LSTM encoder with a hidden size 256. All activation functions in the neural networks are ReLU. The baseline models are optimized with an Adam optimizer with a learning rate of $10^{-3}$, exponentially decayed to $5 \times 10^{-4}$ over 1500 training epochs. Due to the limited time horizon of training data, the baseline models can only fit high-variance labels, leading to underperforming prediction results.

**SPE diffusivity prediction baseline model.** We adopt the GNN model proposed in (Xie et al., 2022) and refer interested readers to (Xie et al., 2022) for more details. The only modification we make is changing the training objective so as to rectify the overestimation coming from the short MD horizon of the training data. We let the baseline model fit the 5 ns diffusivity curves (curves in Figure 9 (f)) by predicting two parameters in the function: $f(t) = y + e^{-\theta t}$. Therefore, the model approximates the converging process of diffusivity with exponential decay. During the evaluation, we set $t = 50$ ns to obtain the model prediction for 50-ns diffusivity. However, without any long training trajectories, it is very challenging to guess the decaying process of diffusivity for various SPEs. The baseline model performs better than using 5-ns ground truth MD but still significantly overestimates particle diffusivity. We have also attempted using a neural ordinary differential equation (Chen et al., 2018) to fit the decay curve but found the results worse than using the parameterized exponential decay function.

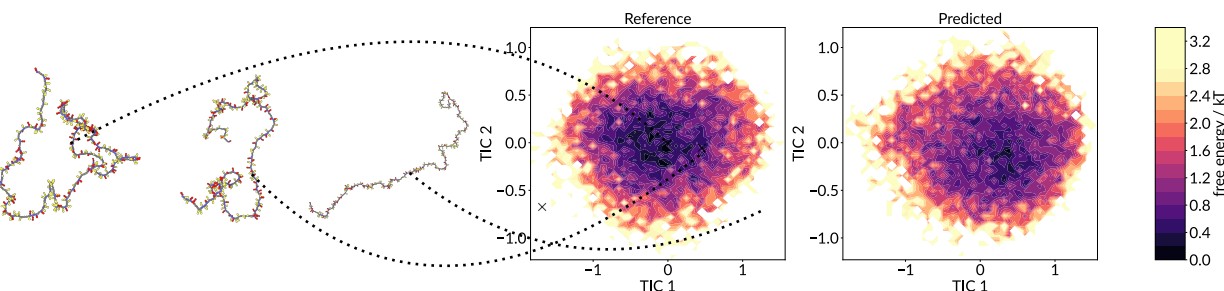

Figure 12: Three representative states with a low, medium, and high radius of gyration (with low to high free energy) are visualized for an example polymer. Free energy surface under a two-dimensional representation is produced with TICA. The projection is fitted over the reference data only and is applied to both reference and learned simulated data.

**Understanding radius of gyration.** The mean squared gyration radius of a polymer is of significant practical importance for understanding the rheological properties of polymers in solution and the compactness of polymers. This is due to its role in establishing a concentration threshold associated with the beginning of chain entanglements and the process of gelation (Webb et al., 2020). Intuitively, the radius of gyration is a way to describe the size of a polymer chain, indicating how spread out the polymer chain is. In Figure 12, we visualize three states with low to high radius of gyration and their location on a two-dimensional free energy

surface produced with time-lagged independent component analysis (TICA, Pérez-Hernández et al. 2013). If the radius of gyration is high, the polymer chain is more spread out, and vice versa.

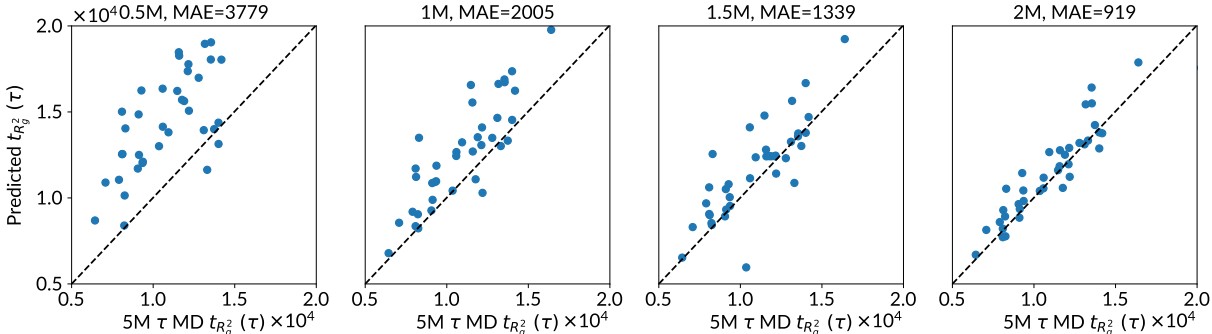

Figure 13: The relaxation time of the squared radius of gyration for the testing polymers estimated using reference trajectories of different lengths. The trajectory length ranges from 0.5M, 1M, 1.5M, and 2M $\tau$ from left to right.

**ACF estimation.** To better understand our model's performance in recovering the ACF of $R_g^2$ for the single-chain polymers, we investigate how well trajectories of different lengths can reconstruct the ACF curves and the relaxation time of $R_g^2$. Figure 13 demonstrates the relaxation time of the squared radius of gyration for the testing polymers estimated using reference trajectories of 0.5M, 1M, 1.5M, and 2M $\tau$ long, along with the average mean absolute error across all 40 testing polymers. Figure 14 demonstrates the corresponding ACF curves for the same four polymers reported in Figure 5. Our training trajectories are 50k $\tau$ long, while the testing trajectories are 5M $\tau$ long. The training trajectories are not long enough for extracting the ACF curves and the relaxation time, so the baseline supervised learning models are not applicable. 5M trajectories simulated with our method result in an MAE of 1657 $\tau$ in estimating the relaxation time (Figure 5), which is between the performance of 1M ground truth trajectory and 1.5M ground truth trajectory.

**Detailed baseline results on estimating the distribution of $R_g^2$.** We include the performance of the supervised learning baseline models on estimating the distribution of $R_g^2$ for the same four example polymers reported in Figure 4 in Figure 15. We also annotate the EMD achieved for each example. Our model can capture the distribution much more faithfully (Figure 4).

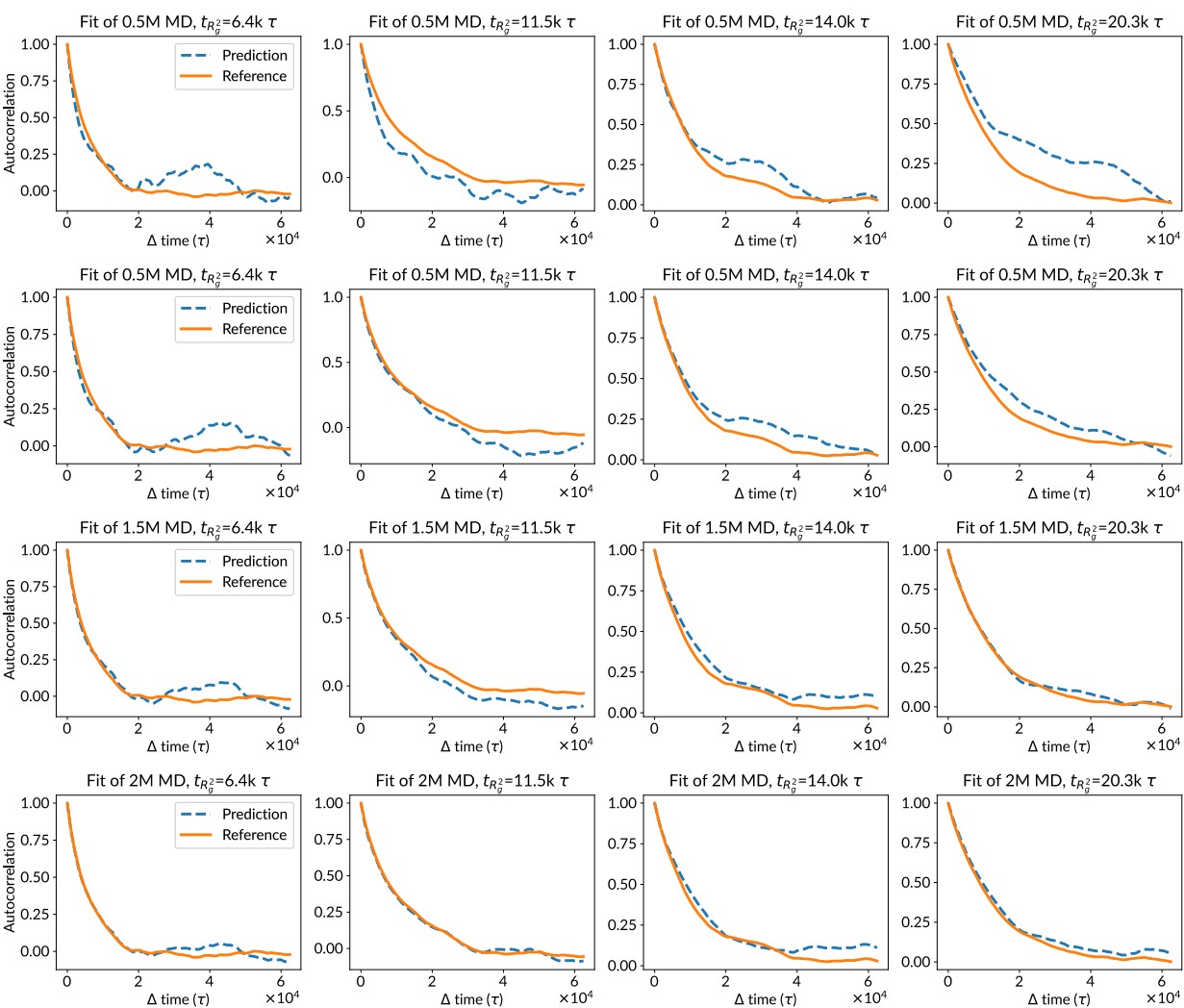

Figure 14: The ACF curves produced by reference trajectories of different lengths for four example polymers. The trajectory length ranges from 0.5M, 1M, 1.5M, and 2M $\tau$ from the first to the fourth row.

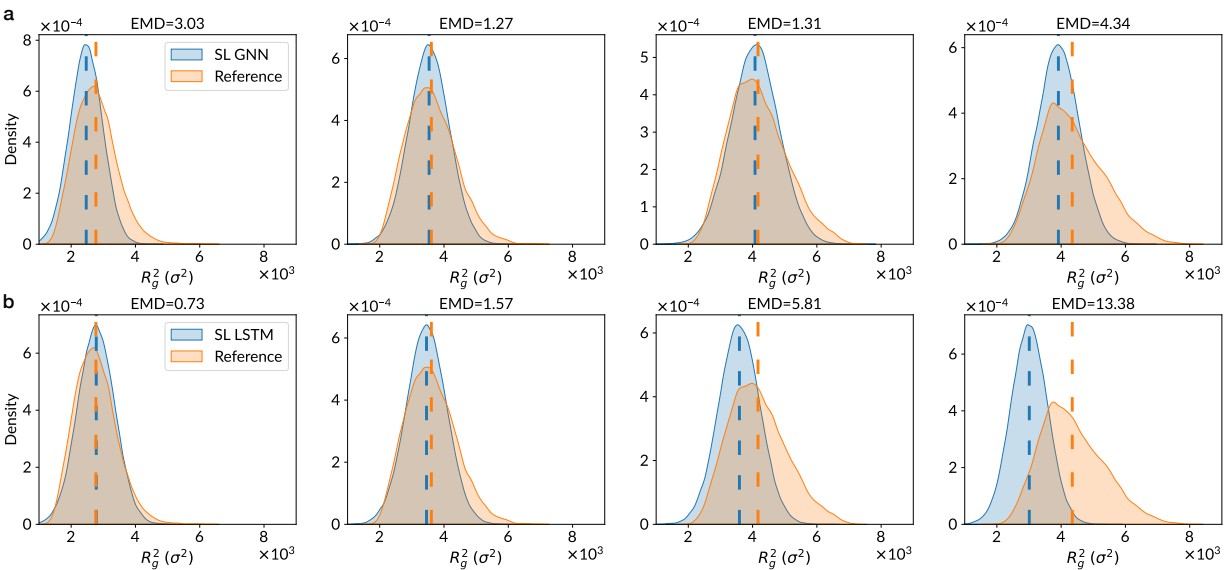

Figure 15: The distribution of $R_g^2$ from the baseline supervised learning models compared to the reference ground truth data for four example polymers with small to large $\langle R_g^2 \rangle$. The first row demonstrates the results from the SL GNN model. The second row demonstrates the results from the SL LSTM model.

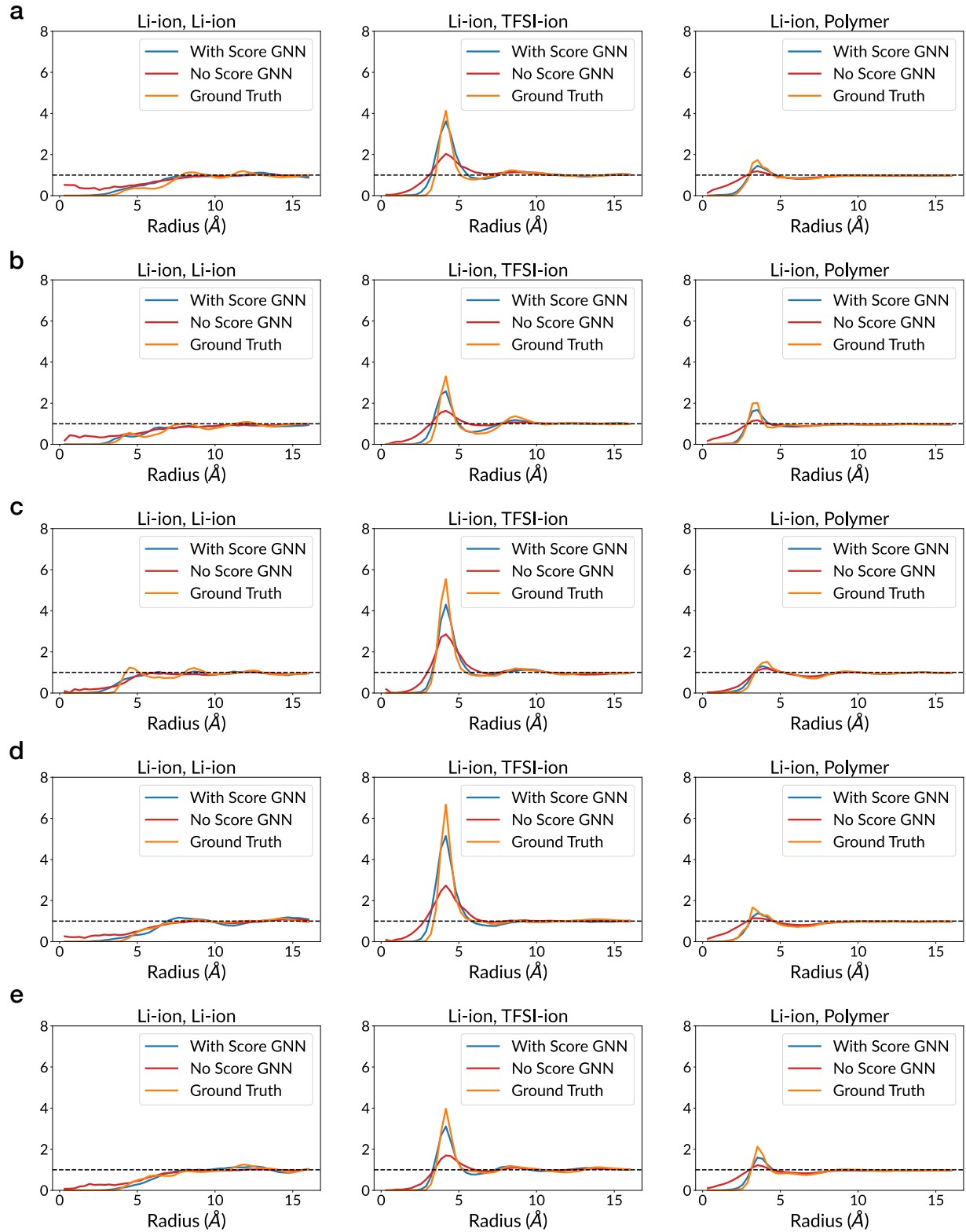

Figure 16: (a,b,c,d,e) Comparison of our model with/without the Score GNN refinement, and the ground truth MD simulation, on RDF of Li-ions (column 1), RDF of Li-ions and TFSI-ions (column 2), and RDF of Li-ions and polymer particles (column 3) for five sampled SPEs.

