# OpenReview forum: "Simulate Time-integrated Coarse-grained Molecular Dynamics with Multi-scale Graph Networks"
_TMLR — Accepted by TMLR_

### Review · Reviewer_bcdg · 2023-06-01

**Summary Of Contributions:**

The paper proposes training coarse-grained GNNs to run molecular dynamics simulations in relatively long time scales. The paper utilizes three GNN models, one for embedding, one for the next time step prediction, and one for the distributional correction. The empirical studies show that the proposed method can achieve stable simulation on molecules of thousands of atoms while giving reasonable property predictions for down-stream tasks.

**Audience:**

Yes

**Claims And Evidence:**

Yes

**Requested Changes:**

Please see the questions in the previous part.

**Strengths And Weaknesses:**

Strengths:
The proposed method can achieve stable simulation at relatively long time scales and generate reasonable property predictions for down-stream tasks.


Weaknesses and Questions:
1. The CG method is based on a graph partitioning algorithm. This is advantageous for its simplicity. However, this can be a potential problem as the graph partitioning algorithm does not account for chemical properties of the molecule. Why is it the case that having roughly equal-sized partition blocks is a desirable property for the partitioning algorithm?


2. The score GNN is not described very clearly. Why such a diffusion based model can help with the stability issue? Or what is the anticipated cause of the stability issue, the inaccurate prediction or the large variance from the dynamics GNN? How is the correction applied precisely during inference (it's briefly mentioned that it follows the annealing Langevin dynamics from previous works)?


3. The overall loss is a direct combination of the two component losses. Are there any attempts to weigh the two terms differently? What about the convergence for both loss terms during training?


4. How is the time step size decided for the experiments? It seems that the time step size is implicitly modeled in the training procedure. For a new application, how would the user know the right time step size to train the model on? Is the model's performance robust to a wide range of various time step sizes?


5. How could this work be combined with other works on studying particular GNN structures for MD related tasks (the paper mentioned that extending the current model to account for more invariant/equivariant properties is a promising direction)? Is it possible to just replace the GNN model structures with other structures with desirable properties? Also, could these properties solve the stability problem? If so, how do we compare this approach to the score correction approach as proposed?


Minor:


What is "q" in equation 1?

---

> ### Author Response · Authors · 2023-06-22
> **Response part 1**
>
> We thank reviewer bcdg for helpful feedback and comments. We address each of the reviewer’s concerns below.
>
> > The CG method is based on a graph partitioning algorithm. This is advantageous for its simplicity. However, this can be a potential problem as the graph partitioning algorithm does not account for chemical properties of the molecule. Why is it the case that having roughly equal-sized partition blocks is a desirable property for the partitioning algorithm?
>
> Thank you for the insightful question. As all coarse-graining procedure risks losing important information for chemical properties, our aim is to preserve as much useful information for simulation as possible. In terms of the coarse-graining mapping, we can either use a predefined procedure like the graph partitioning algorithm presented or use a learned coarse-graining mapping. We choose METIS after experimenting with several possible approaches, including learning the coarse-graining mapping end to end with DiffPool [1]. However, a changing coarse-grained mapping requires the simulator to predict not only the position of the CG beads but also their composition for the next step. This turns out to make the simulation very unstable, as the error accumulation over the composition is hard to correct. The intuition behind the graph partitioning approach is that two atoms grouped into the same CG bead in this procedure will never be far away from each other in the entire simulation since they are tied by a path of bonds. As the CG bead embedding is obtained from the time-invariant bond graph, the CG bead type embedding will not change throughout the entire simulation. This simplification of the prediction task is crucial for stable simulations. A roughly equal-sized partition is desired because it normalizes the distribution of pairwise distances and the magnitude of velocity/acceleration across different CG beads. This normalization effect is highly beneficial to training stability and efficiency. We have included more details of the coarse-graining process in Section 3.
>
> > The score GNN is not described very clearly. Why such a diffusion based model can help with the stability issue? Or what is the anticipated cause of the stability issue, the inaccurate prediction or the large variance from the dynamics GNN? How is the correction applied precisely during inference (it's briefly mentioned that it follows the annealing Langevin dynamics from previous works)?
>
> The main reason for instability is error accumulation in the forward simulation. The diffusion-based corrector can help with stability issues because it is capable of removing all levels of unphysical errors through an iterative refinement process at each forward step to avoid error accumulation. We added more detailed paragraphs describing the motivation and the correction procedure.
>
> > The overall loss is a direct combination of the two component losses. Are there any attempts to weigh the two terms differently? What about the convergence for both loss terms during training?
>
> We did not attempt to weigh the two terms because the two terms do not conflict with each other. Both loss terms stably decrease and plateau as training proceeds.
>
> > How is the time step size decided for the experiments? It seems that the time step size is implicitly modeled in the training procedure. For a new application, how would the user know the right time step size to train the model on? Is the model's performance robust to a wide range of various time step sizes?
>
> The time step size is an important hyperparameter. The consequence of a longer time step includes (1) higher simulation speed; (2) more movements getting smoothed out;  (3) observables at a fine time scale becoming unavailable. For a new application, a user will first decide which observables are of interest and the time scale of these observables. Then conditional on the interested observables being preserved, use the longest possible time step. It will also depend on the dataset, as a longer time step means a smaller amount of training data. In Appendix A, we include an ablation study over the time step size for our Li-ion battery experiment.

---

> > ### Author Response · Authors · 2023-06-22
> > **Response part 2**
> >
> >
> > > How could this work be combined with other works on studying particular GNN structures for MD related tasks (the paper mentioned that extending the current model to account for more invariant/equivariant properties is a promising direction)? Is it possible to just replace the GNN model structures with other structures with desirable properties? Also, could these properties solve the stability problem? If so, how do we compare this approach to the score correction approach as proposed?
> >
> > We believe the adaption of more recent equivariant GNN architecture is a promising next step. Many existing equivariant GNNs developed for applications such as force fields and property prediction cannot be readily used because they only take a single configuration as the input. In our experiments, we observe that in the partially observed coarse-grained settings, conditional on historical information is crucial. We believe that incorporating historical information in an equivariant way by extending equivariant GNN (e.g., NequIP[2]) is a promising future direction. Additionally, equivariant GNNs are usually not computationally efficient as they often involve expensive many-body interaction (e.g., GemNet involves three-body/four-body message passing) or tensor products (e.g., NequIP). More efficient equivariant models (e.g., Allegro [3]) might be necessary for handling large-scale systems and datasets studied in this paper. We have included this discussion on incorporating equivariant GNNs in the paper.
> >
> > We believe the improvement coming from a more accurate dynamics model will be orthogonal to the stability benefit from a score-based corrector. Error accumulation into unphysical configuration is the main cause of simulation instability. Respecting physical symmetry may help to mitigate the prediction error and relieve the stability issue. However, all inputs to a dynamical model will be physical states coming from the training dataset, and the ground truth next step will also be a physical state. Once the error shows up in the bootstrapped state, it is hard for a dynamical model to remove it as the noise was not part of its training. There exist noise augmentation methods, but large noise cannot be used because it injects bias into the training process [4]. The score-based corrector tackles stability from a different perspective. The denoising score-matching training objective explicitly asks the model to denoise different levels of noisy structures to the history-conditional future state, which lies on the manifold of physical states. The iterative refinement procedure is also powerful in correcting errors from an imperfect dynamical model. For challenging simulation tasks, we believe both advanced GNN architecture and the score-based corrector are beneficial for preserving stability.
> >
> > In summary, we believe there are concrete ways to extend existing equivariant GNNs for time-integrated simulators. The explicit denoising objective of a score-based corrector offers a unique advantage in removing accumulated errors in long-time learned simulations.
> >
> > > What is "q" in equation 1?
> >
> > Thank you for pointing out this typo. It should have been "$\mathbf{x}$". We have revised this in the manuscript.
> >
> >
> > Reference:
> >
> > [1] Ying, Zhitao, et al. "Hierarchical graph representation learning with differentiable pooling." Advances in neural information processing systems 31 (2018).
> >
> > [2] Batzner, Simon, et al. "E (3)-equivariant graph neural networks for data-efficient and accurate interatomic potentials." Nature communications 13.1 (2022): 2453.
> >
> > [3] Musaelian, Albert, et al. "Learning local equivariant representations for large-scale atomistic dynamics." Nature Communications 14.1 (2023): 579.
> >
> > [4] Sanchez-Gonzalez, Alvaro, et al. "Learning to simulate complex physics with graph networks." International conference on machine learning. PMLR, 2020.

---

### Review · Reviewer_19Rz · 2023-06-08

**Summary Of Contributions:**

The authors propose a machine learning system to perform long simulations on many-atom systems with chemically-informed multiscale models. The authors have quite robust statistics and experiments with reasonable cross validation to back up their claims.

**Audience:**

Yes

**Claims And Evidence:**

Yes

**Requested Changes:**

It would be great to give the readers some intuition of squared radius of gyration. What does a good or bad value look like, and what are the potential ways bad statistics arise?

For Figure 4, what do those distributions look like for the other ML baselines? Also, are you also able to show the EMD, as you did in Table 2? It’d be great to compare that sense of variability.

Is the only source of instability “bead collision”? Are there other aspects of stability, or non-chaotic systems that could be considered? This seems like an instance of the “exploding gradient problem”: Pascanu, Razvan, Mikolov, Tomas, and Bengio, Yoshua. Understanding the exploding gradient problem. arXiv preprint arXiv:1211.5063, 2012.

I would like a little bit more intuition about Figure 9 in the text. How should we interpret log(cm^2/s), and why is it consistently higher, and not lower, as a predicted value?

With regards to the rotational and translational equivariance described in the discussion, I don’t think that new analysis and methods should be introduced. If it is that important, it should be included into the main work, or otherwise a supplemental note.


**Strengths And Weaknesses:**

Strengths

I think it is interesting for the authors to point out there are good coarse grained model for long time series things for Li-ion polymer electrolytes.

I like that the bead type embeddings is chemically informed, with unique values for atom type, molecular weight, and bond type.

I like how in section 4.1 training was done on class I and then testing was done in class II.

I’m very happy with the Earth Mover’s Distance (EMD) statistic presented in Table 2. I think that is a great way to show and compare different distributions across your models rather than just showing a plot, which is what is typically done.

I think Figure 6c is very helpful! I like how each of the steps of coarse-graining are shown.


Weaknesses


How many k historical CG states are needed? Is it Markovian, and why do we need to know a long state?

Why do we need the Graph-clustering CG model, METIS? This is actually how you go from the atom-represenation to the bead representation? Does this representation change over the course of the simulation? This seems like it would not change much for single, long polymers, but would change a lot for simulations with few bonds. Are the only bonds you are using covalent?

For Figure 3, what does the r^2 tell us? What does the LSTM result mean? The LSTM result seems off relative to others.

Figure 5b is not very useful as a comparison. It seems like the relative distribution of free energy is similar, but the TICA dimensions aren’t meaningful or comparable between Reference and Predicted. Are there any other dimensions or values to plot that would be more useful?

---

> ### Author Response · Authors · 2023-06-22
> **Response part 1**
>
> We thank reviewer 19Rz for helpful feedback and comments. We address each of the reviewer’s concerns below.
>
> > How many k historical CG states are needed? Is it Markovian, and why do we need to know a long state?
>
> Although the ground truth reference all-atom simulation is Markovian, the coarse-grained dynamics is not Markovian as coarse-graining introduces partial observability of the atomic system. The system is only Markovian when the position of every atom is known. However, coarse-graining leads to a partially observed state. Therefore, history dependence becomes important. For such a partially observed system, the required historical state sequence length depends on many factors, such as the inherent complexity of the fine-grained system and the coarse-grained mapping. The history length is a hyperparameter that we conduct an ablation study in Appendix A, where we found a long history conditioning is beneficial for modeling the solid polymer electrolyte systems.
>
> > Why do we need the Graph-clustering CG model, METIS? This is actually how you go from the atom-represenation to the bead representation? Does this representation change over the course of the simulation? This seems like it would not change much for single, long polymers, but would change a lot for simulations with few bonds. Are the only bonds you are using covalent?
>
> Thank you for the insightful question. We use the graph clustering algorithm METIS to go from atomic resolution to bead representation. This representation does NOT change over the course of the simulation. In this work, we study systems in equilibrium where there is no bond breaking or formation. Therefore, the CG mapping based on graph clustering over the bond graph is persistent throughout the entire simulation for an arbitrarily long time. We use all chemical bonds in the graph. Such a fixed CG mapping approach is also employed in many highly successful classical CG force fields, such as MARTINI [1].
>
> The intuition behind this choice is that two atoms grouped into the same CG bead will never be far away from each other since they are tied by a path of bonds. As the CG bead embedding is obtained from the time-invariant bond graph, the CG bead type embedding will not change throughout the entire simulation. This is a design choice that we made after experimenting with several CG approaches, including more flexible CG methods where we allow the (potentially learned with DiffPool [2]) coarse-grained mapping to change over time. However, a changing coarse-grained mapping requires the simulator to predict not only the position of the CG beads but also their composition for the next step. This turns out to make simulation very unstable, where error accumulation over the composition leads to the breakdown of simulations.
>
> > For Figure 3, what does the r^2 tell us? What does the LSTM result mean? The LSTM result seems off relative to others.
>
> The r2 score is the coefficient of determination that evaluates the goodness-of-fit of different models in predicting the mean squared radius of gyration for the test dataset. The LSTM model attains a negative r2, indicating bad performance. Since we are evaluating the learned simulator on a property prediction task, we establish two baseline models that predict the mean squared radius of gyration from sequence-level information (LSTM) and graph-level information (GNN). The LSTM model does not perform well due to incomplete information from sequence-level representation.
>
> We note that our task is very challenging, where only short trajectories (50k $\tau$) are used for training that can give very noisy labels for the prediction task (where labels are from 5M $\tau$ reference trajectories). The learned simulator is able to overcome this challenge because by learning to simulate, we can bootstrap to longer timescales.
>
> > Figure 5b is not very useful as a comparison. It seems like the relative distribution of free energy is similar, but the TICA dimensions aren’t meaningful or comparable between Reference and Predicted. Are there any other dimensions or values to plot that would be more useful?
>
> Thank you for your valuable suggestion. We appreciate your input regarding the use of TICA in our work. The main motivation behind incorporating TICA into our study was to provide a more detailed analysis of the structural information retained in the learned simulation. We agree with your point that TICA is more informative for systems with complex free energy surfaces where multiple metastable states exist. To address your suggestion and provide a more comprehensive analysis, we have included a contact map visualization of both the reference trajectory and the learned simulated trajectory in Fig 5. We hope this visualization offers a clearer representation of the recovery of various levels of structural information.

---

> > ### Author Response · Authors · 2023-06-22
> > **Response part 2**
> >
> > > It would be great to give the readers some intuition of squared radius of gyration. What does a good or bad value look like, and what are the potential ways bad statistics arise?
> >
> > Thank you for the great suggestion. We have added more explanation and visualization to Appendix C and Fig 12 to understand the squared radius of gyration. The squared radius of gyration can be used as a measure of the compactness of the polymer. A polymer with a lower squared radius of gyration has a more entangled 3D structure ensemble, while a higher squared radius of gyration indicates a more untangled structure ensemble. Stronger interactions between the beads lead to a more tangled structure, which has a lower squared radius of gyration. Bad statistics arise when a model mispredicts the interaction between the polymer beads.
> >
> > > For Figure 4, what do those distributions look like for the other ML baselines? Also, are you also able to show the EMD, as you did in Table 2? It’d be great to compare that sense of variability.
> >
> > We have included the distribution predicted by the baseline models in Appendix C, Fig 15. We have also included the EMD values for each example.
> >
> > > Is the only source of instability “bead collision”? Are there other aspects of stability, or non-chaotic systems that could be considered? This seems like an instance of the “exploding gradient problem”: Pascanu, Razvan, Mikolov, Tomas, and Bengio, Yoshua. Understanding the exploding gradient problem. arXiv preprint arXiv:1211.5063, 2012.
> >
> > Stability is a common challenge in learning to simulate. The main source of instability is error accumulation, while “bead collision” is a consequence of such error accumulation that can be readily observed from the RDF curves. Therefore we use it as a metric to compare models’ stability. Given the complexity of atomic systems, any practical-relevant molecular dynamics will be chaotic. Another aspect of research in stability is to actively collect new data for which the model makes mistakes [3]. We have included this direction in the discussion section.
> >
> > > I would like a little bit more intuition about Figure 9 in the text. How should we interpret log(cm^2/s), and why is it consistently higher, and not lower, as a predicted value?
> >
> > Figure 9 presents the result of the model’s performance in predicting the diffusivity of Li ions and TFSI ions. When we talk about diffusivity in the context of a Li-ion solid polymer electrolyte, we're essentially referring to how fast ions can move through the electrolyte. The movement of these ions is important because it enables the flow of electric charge in batteries and other energy storage devices. The unit of diffusivity is cm^2/s. Often diffusivity is reported in log scale [3] due to the large variance of diffusivity across different polymers.
> > When the MD simulations are not long enough, we may observe that the ions can move in a direct and unhindered way. It occurs over short distances and is characterized by fast and uninterrupted ion motion. With a long enough simulation, The ions will interact with the surrounding material and other ions, causing them to change direction frequently. The simulation is said to “enter the diffusive regime” with a long enough simulation time, where the movement of ions becomes random and slow, like a ball bouncing around in a pinball machine, changing direction with each collision. Therefore, ion diffusivity is a quantity that converges with a longer simulation time, while a short simulation would overestimate diffusivity.
> >
> > Note that in our experiment, the training trajectories (5 ns) are much shorter than the time required (50 ns) for estimating ion diffusivity reliably (Fig 7). The first column of Fig 9 demonstrates the overestimation from directly using 5-ns trajectories to compute diffusivity. For the supervised learning model, we used a curve-fitting approach described in the Appendix C to tackle the overestimation problem but found it to be very challenging. For the learned simulator without score-based correction, the unstable dynamics lead to faster motion of the ions and thus overestimation of the diffusivity. Our full model was the best performing model as it is capable of learning the dynamics from short trajectories and bootstrapping long trajectories stably.
> >
> > > With regards to the rotational and translational equivariance described in the discussion, I don’t think that new analysis and methods should be introduced. If it is that important, it should be included into the main work, or otherwise a supplemental note.
> >
> > Thank you for the suggestion; we have extended and moved the discussion on equivariance to the supplementary materials.

---

> > > ### Author Response · Authors · 2023-06-22
> > > **Response part 3**
> > >
> > > Reference:
> > >
> > > [1] Marrink, Siewert J., et al. "The MARTINI force field: coarse grained model for biomolecular simulations." The journal of physical chemistry B 111.27 (2007): 7812-7824.
> > >
> > > [2] Ying, Zhitao, et al. "Hierarchical graph representation learning with differentiable pooling." Advances in neural information processing systems 31 (2018).
> > >
> > > [3] Vandermause, Jonathan, et al. "On-the-fly active learning of interpretable Bayesian force fields for atomistic rare events." npj Computational Materials 6.1 (2020): 20.
> > >
> > > [4] Xie, Tian, et al. "Accelerating amorphous polymer electrolyte screening by learning to reduce errors in molecular dynamics simulated properties." Nature communications 13.1 (2022): 3415.

---

### Review · Reviewer_wvWc · 2023-06-15

**Summary Of Contributions:**

This paper introduces a new machine learning pipeline that emulates molecular dynamics computations. The overall procedure is a multi-stage process:

1. A GNN operating on the atomic structure learns local per-atom and per-edge features
2. A "coarse graining" operation is applied, which agglomerates atom features into features which represent clusters ("beads") in the original graph, and connects them with corresponding edges
3. A "dynamics network" predicts the motion of these coarse-grained beads at $\Delta t$ in the future given current and recent positions and embeddings
4. The predictions are refined using a score-based denoising diffusion model to generate the final predicted locations of the coarse-grained beads

The atom features and the dynamics network are trained end-to-end; the refinement process is trained following a denoising score-matching objective.

A key point in the efficiency of the learned model is that $\Delta t$ from the machine learning model is a much longer timestep than one forward step of a traditional molecular dynamics simulator.

There are a good set of experiments which successfully defend the approach, showing good generalization to much longer simulations than the training regime, and including good ablation studies. Stability of the approach under long iterations is decent, and clearly improved by the use of the diffusion refinement.

**Audience:**

Yes

**Broader Impact Concerns:**

No concerns

**Claims And Evidence:**

Yes

**Requested Changes:**

My requested changes are largely minor.

Regarding confusion on the coarse-graining process:

* How are fine-grained atoms assigned values M? Page 4 includes a reference to a function $C(\mathbf{v}^F_{i,t})$ which computes the corresponding assignment $m \in { 1,\dots,M \}$. However, I don't understand where this function comes from. Is this fixed ahead of time by the practitioner? If so, how should this be done in general? Or is it a trained neural network which acts as a clustering model? This part of the paper was not clear to me.
* Is this coarse-graining process stable over time? For example, if you computed CG beads at time $t_0$, then ran MD forward some duration $\Delta t$ and computed CG beads again, would the resulting CG graph (i.e. in terms of assignments of atoms to beads, and in definition of edges) be the same at both $t_0$ and $t_0 + \Delta t$? Unless I am missing something this would not in general be true. Maybe this does not matter, but still (along with the previous bullet point) this contributed to some confusion around the CG process.
* In section 4.1 the coarse-grained beads are further coarse-grained. Are we okay with this? Why? This might not need clarified in the paper itself, a comment to me is fine.

Regarding end-to-end training:

* Why is the score-based diffusion trained on a general denoising objective with Gaussian noise injected, and then used to specifically "denoise" the output of the dynamics network? It seems to me that this would be a reasonable choice only if the errors from the dynamics network were themselves Gaussian and of a particular scale. If the error distribution of the dynamics network were somehow biased then I would think that instead the dynamics network itself should be used as a source of "noise". In any case I would like to hear the motivation for this (even if it does not make it to the main paper), as this choice seemed odd.

Regarding presentation:

* For figure 2(d) isn't the left figure the original fine-grained polymer? It is labeled as the coarse-grained one. Or is this the output of the first intermediate coarse-graining? This is unclear.
* Something is wrong with the $r^2$ values for the LSTM in Fig 3 and Table 2. In Fig 3 the value is larger than the method marked "Ours", despite (visually) seeming to have a much lower correlation. In table 2, the number for $r^2$ for the LSTM is negative, which is impossible (unless this is "adjusted" in some way that I didn't catch).

Regarding baselines:

* As mentioned in "negatives", in e.g. the ACF computation in Figure 5(a): I see that this broadly tracks the reference, but is this performance "good" or "excellent"? By what standards? How would baselines perform here — surely there must be some meaningful baselines, even if long-run MD computations are not feasible. As it stands the paper claims this is "excellent" performance but the figures are lacking baselines to demonstrate this adequately. (Even including the supervised learning methods, which I believe could be applied here but would suffer due to the need for long extrapolations, would be helpful….)



**Strengths And Weaknesses:**

Strengths:

* Model is fairly approachable and understandable to non-experts in molecular dynamics (e.g. to a general machine learning audience), as is the paper presentation.
* Performance on computing aggregated statistics from long-running simulations is impressive

Weaknesses:

* For some of the experiments, I don't have a good sense of the baseline performance. This is particularly the case in e.g. the ACF computation in Figure 5(a): I see that this broadly tracks the reference, but is this performance "good" or "excellent"? By what standards? How would baselines perform here — surely there must be some meaningful baselines, even if long-run MD computations are not feasible. As it stands the paper claims this is "excellent" performance but the figures are lacking baselines to demonstrate this adequately. (Even including the supervised learning methods, which I believe could be applied here but would suffer due to the need for long extrapolations, would be helpful….)
* I had some confusion regarding how the coarse grain bead assignments of individual atoms are performed, and to what extend the assignment would be stable over time for large simulations. This is particularly confusing to me as the CG beads are themselves further coarsened at a later stage. More details on this in "requested changes".
* It's not obvious to me that MD computations at this coarse-grained level are usable for most of the tasks for which someone may be interested in an MD simulator. In particular, unless I misunderstand it is not possible to recover coordinates of the original "fine grained" atoms. As such it would be nice if some convincing could be done that this degree of "coarsening" still yields a useful general-purpose tool (say, for predicting protein-ligand binding or unbinding from a transition state, or other such tasks).

---

> ### Author Response · Authors · 2023-06-22
> **Response part 1**
>
> We thank reviewer wvWc for helpful feedback and comments. We address each of the reviewer’s concerns below.
>
> > For some of the experiments, I don't have a good sense of the baseline performance. This is particularly the case in e.g. the ACF computation in Figure 5(a): I see that this broadly tracks the reference, but is this performance "good" or "excellent"? By what standards? How would baselines perform here — surely there must be some meaningful baselines, even if long-run MD computations are not feasible. As it stands the paper claims this is "excellent" performance but the figures are lacking baselines to demonstrate this adequately. (Even including the supervised learning methods, which I believe could be applied here but would suffer due to the need for long extrapolations, would be helpful….)
>
> Thank you for pointing out the unclarity in our ACF results. The baseline models are not applied to the ACF task because computing ACF requires trajectories much longer than training trajectories. To give meaningful baselines, we include further results on attempting to estimate the ACF with shorter trajectories than the evaluation trajectories.
>
> Our training trajectories are 50k $\tau$ long, while the testing trajectories are 5M $\tau$ long. We use a series of different portions of the reference trajectories from short to long (the first 0.5M, 1M, 1.5M, 2M $\tau$ of the 5M $\tau$ reference trajectories) to investigate what accuracy can the ACFs and relaxation time scale be captured with shorter trajectories. We add Fig 13, where we plot the regression performance over the relaxation time for the estimation from four different trajectory lengths and report the MAE over relaxation time. Our model achieves an MAE of 1657 $\tau$, which is between the performance of estimation using 1M $\tau$ and estimation using 1.5M $\tau$. We further report the ACF curves obtained from 0.5M, 1M, 1.5M, and 2M $\tau$ trajectories for the four example polymers by adding Fig 14.
>
> We note that the estimation of ACF and the relaxation time is a very difficult task. To obtain the relaxation time, we use least-square regression (https://docs.scipy.org/doc/scipy/reference/generated/scipy.optimize.curve_fit.html) to fit the ACF curves. However, when the trajectory length is too short, curve fitting will throw numerical errors. In our experiment, curve fitting cannot be accomplished due to numerical instability when the reference trajectory is less than 0.5M $\tau$ long. Note that our training trajectories are only 50k $\tau$, so the ACF and relaxation time results cannot be obtained from our training data or the supervised learning models (which only have access to the training data).
>
> > I had some confusion regarding how the coarse grain bead assignments of individual atoms are performed, and to what extend the assignment would be stable over time for large simulations. This is particularly confusing to me as the CG beads are themselves further coarsened at a later stage. More details on this in "requested changes".
>
> We are sorry for the confusion. We added more details on the coarse-graining algorithm to the paper. The coarse-graining assignment is consistent throughout the simulation, as the graph clustering is performed over the bond graph, which is time-invariant in our experiments.
>
> The CG beads are NOT further coarsened at a later stage of our method. For all our experiments, the coarse-graining procedure is only applied once. In the single-chain polymer example, the reference data itself is defined as a coarse-grained system (instead of an atomic system) with associated force terms to begin with.
>
> In summary, our method involves a one-time coarse-graining step using a graph clustering algorithm. For the single-chain polymer dataset, the inputs are already in a coarse-grained representation, which is unrelated to the coarse-graining performed by our method. Further information about the single-chain polymer dataset can be found in [1]. We have changed our wording in several paragraphs to avoid ambiguity. We understand that this explanation may still be confusing, and we are more than happy to provide additional clarification if needed.

---

> > ### Author Response · Authors · 2023-06-22
> > **Response part 2**
> >
> >
> > > It's not obvious to me that MD computations at this coarse-grained level are usable for most of the tasks for which someone may be interested in an MD simulator. In particular, unless I misunderstand it is not possible to recover coordinates of the original "fine grained" atoms.
> >
> > In this paper, we focus on enabling stable and accurate simulation at the coarse-grained level and demonstrate useful observables that can be recovered from the coarse-level simulations. Coarse-level simulations are widely studied and used for simulating large-scale systems that are too expensive to simulate in full-atom resolution. For example, polymers, membranes, proteins, and condensates [2,3,4,5]. Our experiments demonstrate the proposed model can accurately recover properties of interest for single-chain polymers and solid polymer electrolytes.
> >
> > On the other hand, backmapping to the fine-grained level is an active research topic and has been the focus of some recent works [6]. Combining the coarse-grained simulation and backmapping approach to simulate at coarse and obtain fine-grained trajectories will be an interesting avenue for future work.
> >
> > > How are fine-grained atoms assigned values M?
> >
> > The assignment is through the graph partitioning algorithm METIS [7] over the bond graph. In this way, only atoms connected with a path of bonds will be grouped into the same CG bead. This ensures that atoms within the same CG bead are always not too far from each other in arbitrarily long-time simulations. We have included more details on the graph partitioning algorithm and coarse-graining procedure in Section 3.
> >
> > > Is this coarse-graining process stable over time?
> >
> > Yes, the graph partitioning algorithm is done over the bond graph, and the bonds are time-invariant. The coarse-grained mapping does not change through the course of the simulation. This is a conscious choice after trying out several different coarse-graining approaches, where more flexible coarse-graining turns out to be unstable when simulating long trajectories, especially because a changing coarse-graining mapping would require the forward prediction of the bead composition.
> >
> > > In section 4.1 the coarse-grained beads are further coarse-grained. Are we okay with this? Why? This might not need clarified in the paper itself, a comment to me is fine.
> >
> > We are sorry for the confusion. In section 4.1, the reference single-chain polymers are already “coarse-grained polymers” in the classical force field definition, so the reference data itself is not at full-atom resolution. Our method only apply coarse-graining once, on top of this dataset that is already “coarse-grained” to begin with.

---

> > > ### Author Response · Authors · 2023-06-22
> > > **Response part 3**
> > >
> > > > Why is the score-based diffusion trained on a general denoising objective with Gaussian noise injected, and then used to specifically "denoise" the output of the dynamics network? It seems to me that this would be a reasonable choice only if the errors from the dynamics network were themselves Gaussian and of a particular scale. If the error distribution of the dynamics network were somehow biased then I would think that instead the dynamics network itself should be used as a source of "noise". In any case I would like to hear the motivation for this (even if it does not make it to the main paper), as this choice seemed odd.
> > >
> > > Thank you for the insightful question. We first note that the diffusion model is conditional on the embedding of historical information. Assuming sufficient training data and an expressive enough model, the optimal score predicted by the score-based corrector will match the score: $\nabla \log p(x_{t+\Delta t} | \mathrm{history})$ at the final noise level. Our dynamics model predicts the time-integrated acceleration as an independent Gaussian distribution with independent Gaussian uncertainty for each particle. Integrating this acceleration results in a next-step prediction with additive independent Gaussian noise. The noise levels and magnitude of the score-based model can be adjusted according to the level of noise present in the dynamics model's predictions, ensuring efficient sampling.
> > >
> > > Furthermore, the use of multi-level Gaussian noise allows for an iterative refinement procedure where the diffusion model can progressively denoise the structure. By applying denoising steps from high to low noise levels, the score-based corrector can fix all levels of incorrect configuration. This iterative refinement is crucial in removing unphysical errors and guiding the model toward high-density areas in the target distribution $p(x_{t+\Delta t} | \mathrm{history})$. In particular, any unphysical local structures will have a very low density in the final level distribution of the score-based corrector. If we were to denoise from a configuration predicted by the dynamics model for training, it would be challenging to incorporate a multi-step and multi-level refinement procedure as we will need to define the forward diffusion process that goes from $p(x_{t + \Delta t} | \mathrm{history})$ to $\hat{p}(x_{t + \Delta t} | \mathrm{history})$, where $\hat{p}$ is predicted by the dynamics model. Recent advancements in distribution-to-distribution diffusion models [8] offer promising insights into this direction and could be explored in future work.
> > >
> > > > For figure 2(d) isn't the left figure the original fine-grained polymer? It is labeled as the coarse-grained one. Or is this the output of the first intermediate coarse-graining? This is unclear.
> > > We are sorry for the confusion. In section 4.1, the reference single-chain polymers are already “coarse-grained” (rather than full-atom) in the classical force field definition. The reference data itself is not at full-atom resolution. Our method only applies coarse-graining once, on top of this dataset that is already “coarse-grained” to begin with.
> > >
> > > > Something is wrong with the r2 values for the LSTM in Fig 3 and Table 2. In Fig 3 the value is larger than the method marked "Ours", despite (visually) seeming to have a much lower correlation. In table 2, the number for r2 for the LSTM is negative, which is impossible (unless this is "adjusted" in some way that I didn't catch).
> > >
> > > Thank you for the question. With the definition that we are using (sklearn.metrics.r2_score https://scikit-learn.org/stable/modules/generated/sklearn.metrics.r2_score.html), r2 can indeed be negative. In both Fig 3 and Table 2, r2 for the LSTM model is -0.91. A negative r2 arises when the model's predictions are worse than a constant function that always predicts the mean of the data. Possible reasons for this performance include (1) short training trajectories giving noisy labels; (2) limited information from sequence representation of the polymers; (3) distribution shift for the training data (class I) to evaluation data (class II).

---

> > > > ### Author Response · Authors · 2023-06-26
> > > > **Response part 4**
> > > >
> > > >
> > > > Reference:
> > > >
> > > > [1] Webb, Michael A., et al. "Targeted sequence design within the coarse-grained polymer genome." Science advances 6.43 (2020): eabc6216.
> > > >
> > > > [2] Kempfer, Kévin, et al. "Development of coarse-grained models for polymers by trajectory matching." ACS omega 4.3 (2019): 5955-5967.
> > > >
> > > > [3] Kmiecik, Sebastian, et al. "Coarse-grained protein models and their applications." Chemical reviews 116.14 (2016): 7898-7936.
> > > >
> > > > [4] Menichetti, Roberto, Kiran H. Kanekal, and Tristan Bereau. "Drug–membrane permeability across chemical space." ACS central science 5.2 (2019): 290-298.
> > > >
> > > > [5] Benayad, Zakarya, et al. "Simulation of FUS protein condensates with an adapted coarse-grained model." Journal of Chemical Theory and Computation 17.1 (2020): 525-537.
> > > >
> > > > [6] Wang, Wujie, et al. "Generative coarse-graining of molecular conformations." arXiv preprint arXiv:2201.12176 (2022).
> > > >
> > > > [7] Karypis, George, and Vipin Kumar. "A fast and high quality multilevel scheme for partitioning irregular graphs." SIAM Journal on scientific Computing 20.1 (1998): 359-392.
> > > >
> > > > [8] De Bortoli, Valentin, et al. "Diffusion Schrödinger bridge with applications to score-based generative modeling." Advances in Neural Information Processing Systems 34 (2021): 17695-17709.

---

### Decision · Action_Editors · 2023-08-04

**Recommendation:** Accept as is

**Comment:**

This paper presents a method for simulating molecular dynamics using graph neural networks, which is otherwise very computationally expensive.  The reviewers all agreed on acceptance, noting that the paper is novel, clear, well-composed and addresses an interesting and important problem.  One reviewer noted that "the paper provides an excellent jumping-off point for further research".  The reviewers also noted that although there were some initial concerns regarding clarity and experiments, the authors did a good job of addressing these in their response.

**Audience:**

This paper has innovations on graph neural networks and applications to modeling molecules and molecular dynamics.  Thus both the methodological modeling community and the application area might be excited about the work.

**Claims And Evidence:**

The reviewers all agreed claims are supported by evidence.